palaeontology/biomechanics/evolution

suspension feeding, *Titanichthys*, Arthrodira, Devonian, comparative biomechanics

**Author for correspondence:**
Samuel J. Coatham
e-mail: sam.coatham@postgrad.manchester.ac.uk

[†]Present address: Department of Earth and Environmental Sciences, Michael Smith Building, University of Manchester, Dover Street, Manchester, UK.

# Was the Devonian placoderm *Titanichthys* a suspension feeder?

Samuel J. Coatham[1,†], Jakob Vinther[1], Emily J. Rayfield[1] and Christian Klug[2]

[1]Life Sciences Building, University of Bristol, 24 Tyndall Avenue, Bristol, UK
[2]Paläontologisches Institut und Museum, Universität Zürich, Karl-Schmid-Strasse 4, 8006 Zürich, Switzerland

SJC, 0000-0003-4597-6210; JV, 0000-0002-3584-9616; CK, 0000-0002-4099-7453

Large nektonic suspension feeders have evolved multiple times. The apparent trend among apex predators for some evolving into feeding on small zooplankton is of interest for understanding the associated shifts in anatomy and behaviour, while the spatial and temporal distribution gives clues to an inherent relationship with ocean primary productivity and how past and future perturbations to these may impact on the different tiers of the food web. The evolution of large nektonic suspension feeders—'gentle giants'—occurred four times among chondrichthyan fishes (e.g. whale sharks, basking sharks and manta rays), as well as in baleen whales (mysticetes), the Mesozoic pachycormid fishes and at least twice in radiodontan stem group arthropods (Anomalocaridids) during the Cambrian explosion. The Late Devonian placoderm *Titanichthys* has tentatively been considered to have been a megaplanktivore, primarily due to its gigantic size and narrow, edentulous jaws while no suspension-feeding apparatus have ever been reported. Here, the potential for microphagy and other feeding behaviours in *Titanichthys* is assessed via a comparative study of jaw mechanics in *Titanichthys* and other placoderms with presumably differing feeding habits (macrophagy and durophagy). Finite-element models of the lower jaws of *Titanichthys termieri* in comparison to *Dunkleosteus terrelli* and *Tafilalichthys lavocati* reveal considerably less resistance to von Mises stress in this taxon. Comparisons with a selection of large-bodied extant taxa of similar ecological diversity reveal similar disparities in jaw stress resistance. Our results, therefore, conform to the hypothesis that *Titanichthys* was a suspension feeder with jaws ill-suited for biting and crushing but well suited for gaping ram feeding.

# 1. Introduction

Some of the largest organisms ever to have roamed the ocean and alive today are suspension feeders. The switch to feeding on the lowest levels of the trophic pyramid is a tremendous shift in food resource [1]. While pursuing large-bodied prey results in adaptations towards stealth, complex hunting behaviours and expanded sensory repertoires, suspension feeding results in a host of anatomical, migratory and behavioural modifications. Locomotory speed and energy reserves scale with body mass—enabling a migratory lifestyle in some species [2] to capitalize on seasonal periods of high food abundance [3]. Invertebrate suspension feeders are known from the Cambrian [4], giant-bodied relative to their temporal counterparts. While the first definitive vertebrate megaplanktivores occurred in the Mesozoic, within the pachycormids [5], this ecological niche may in fact have originated in the Devonian.

The arthrodire *Titanichthys* occurred in the Famennian [6], the uppermost stage of the Devonian (372–359 Ma [7]). There are multiple morphological features indicating that *Titanichthys* may have been a megaplanktivore, primarily its massive size [8]. The elongate, narrow jaws lack any form of dentition or shearing surface [9]; seemingly ill-equipped for any form of prey consumption more demanding than simply funnelling prey-laden water into the oral cavity. *Titanichthys* is also known for its small orbits (in relation to skull size) [8], indicating that visual acuity may not have been that important in its predatory behaviour. This is a known feature of predation in extant suspension feeders [10], so may be further evidence of planktivory. However, the suspension-feeding pachycormid *Rhinconichthys* has enlarged sclerotic rings [11], bringing into question the use of reduced orbitals as a diagnostic character of planktivory.

Despite the numerous physical traits shared between *Titanichthys* and other definitive giant suspension feeders, planktivory in *Titanichthys* has yet to be strongly supported, due to the absence of evidence of a suspension-feeding structure. If *Titanichthys* was indeed a suspension feeder, presumably it would have fed in a roughly analogous manner to modern planktivorous fish, which separate prey from water entering the oral cavity using elaborate or ornamented gill rakers (this was also the suspension-feeding method of planktivorous pachycormids [12]). Placoderm gill arches are rarely preserved [13], so the absence of a fossil suspension-feeding structure may be an artefact of the poor fossil record [14], or it may indicate that *Titanichthys* was not a suspension feeder.

The viability of suspension feeding in *Titanichthys* is promoted by seemingly favourable conditions in the Devonian. Increases in primary productivity appear to be associated with the recurrent evolution of megaplanktivores, with potential expansions of available food resources enabling larger body sizes. This has been observed in the diversification of mysticetes [15,16] and the origin of most suspension-feeding elasmobranch clades [17], with potential further correlations in the evolution of giant planktivorous anomalocarids in the Lower Cambrian [18] and pachycormids in the Jurassic [19]. Productivity probably also increased throughout the Devonian, with the combination of tracheophyte proliferation [20] and the advent of arborescence [21] probably accelerating the rate of chemical weathering [22]. This could have resulted in enrichment of the oceanic nutrient supply via runoff [23], potentially increasing marine productivity [19]. There is little direct proof of this [24], as is typical when trying to track primary productivity [25]. However, we can infer from the rise in diversity of predators with high energetic demands [26] that there was probably sufficient productivity to support relatively complex ecosystems. Consequently, it seems probable that productivity did increase, potentially facilitating the evolution of a giant suspension feeder in the Devonian.

To assess whether *Titanichthys* was indeed a suspension feeder, we investigated the mechanical properties of its jaw in order to infer function. The engineering technique finite-element analysis (FEA) has previously been used to effectively differentiate between the mandibles of related species with differing diets [27]. Consequently, finite-element models of the inferognathals of *Titanichthys termieri*, *Tafilalichthys lavocati* and *Dunkleosteus terrelli* were generated and compared. *Tafilalichthys* is thought to have been durophagous (specialized to consume hard-shelled prey) [28], while *Dunkleosteus* was almost certainly an apex predator [29]; representing the two most plausible feeding modes for *Titanichthys* (excluding planktivory). Both species were arthrodires related to *Titanichthys*, with *Tafilalichthys* more closely related—probably within the same family [8].

By digitally discretizing a structure into many elements and applying loads, constraints and material properties, the stress and strain experienced within each element can be calculated in FEA [30]. When viewed as components of the entire structure, its resistance to stress and strain can be clearly visualized, enabling functional inference. While the magnitude of stress/strain values in extinct taxa are hard to definitively ascertain, comparing between models loaded in the same manner is effective

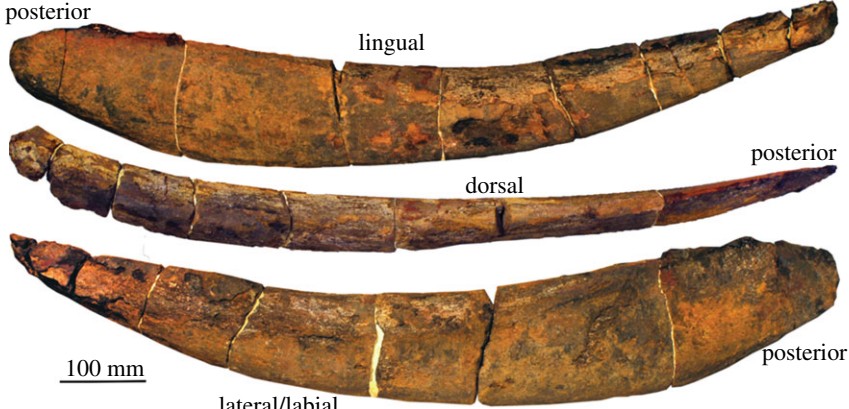

**Figure 1.** Left inferognathal of *Ti. termieri* (PIMUZ A/I 4716), from the Southern Maïder basin, Morocco. The specimen is nearly complete, excluding the anteriormost tip. The inferognathal lacks both dentition and shearing surfaces. It has been glued together where fractures occurred. Photographed at the University of Zurich. Total length = 96 cm.

for comparative studies of function [31]. Therefore, the mechanics of the arthrodire inferognathals will be compared based purely on their shape. Extant taxa, the lifestyles of which are far better understood, will be used as a further reference point, to validate the use of jaw robustness as a proxy for feeding strategy. The sharks *Cetorhinus maximus* (basking), *Carcharodon carcharias* (great white) and *Heterodontus francisci* (horn) all occupy ecological niches roughly analogous to those of the placoderms studied (planktivore, apex predator and durophage, respectively). In addition, the cetaceans *Balaenoptera musculus* (blue whale) and *Orcinus orca* (killer whale) will serve as a further planktivore–apex predator reference; albeit with much greater evolutionary distance between the species.

Comparing the jaw mechanics of definitive suspension feeders with their macrophagous relatives will provide clarity regarding the implications of any differences in stress/strain patterns of the placoderm jaws, informing any conclusions regarding *Titanichthys*' feeding strategy. Should *Titanichthys* have been a suspension feeder, its jaw would be expected to be less mechanically robust than those of related species with diets associated with greater bite forces, which would exert more stress on the jaw. Consequently, the jaw of a suspension feeder is predicted to be less resistant to stress and strain than those of the compared durophagous and macropredatory species.

# 2. Material and methods

## 2.1. Placoderm specimens

*Titanichthys* specimens are mostly known from the Cleveland Shale, with remains of five different *Titanichthys* species having been found there—albeit mostly from relatively incomplete specimens [9]. There have also been species described from Poland and, most pertinently for this study, Morocco. *Titanichthys termieri*, one of the largest members of the genus, is known from the Tafilalet basin in South Morocco [32].

The *Titanichthys* and *Tafilalichthys* specimens used in this study were found in Morocco, where the Famennian strata are known for their high quantity of preserved placoderms [33,34]. Both specimens were discovered in the Southern Maïder basin, which neighbours the Tafilalet basin. The type specimens of both *Ti. termieri* and *Ta. lavocati* were described in the Tafilalet basin [9]; therefore, the fossils in this study can be assigned to those species with some confidence.

The primary subject of this investigation was a nearly complete *Ti. termieri* left inferognathal (PIMUZ A/I 4716—figure 1). It is missing only the anterior tip, representing a small portion of the overall length—with a total length of 96 cm without the tip. While arthrodire inferognathals are typically divided antero-posteriorly into distinct biting and non-biting divisions [35], in *Ti. termieri*, there is a much more gradual transition between the narrow posterior division and the thicker anterior biting division. The posterior blade is narrow mediolaterally and high dorsoventrally, similar to other arthrodires [36]. There are no denticles or shearing surfaces visible on the anterior biting division—a

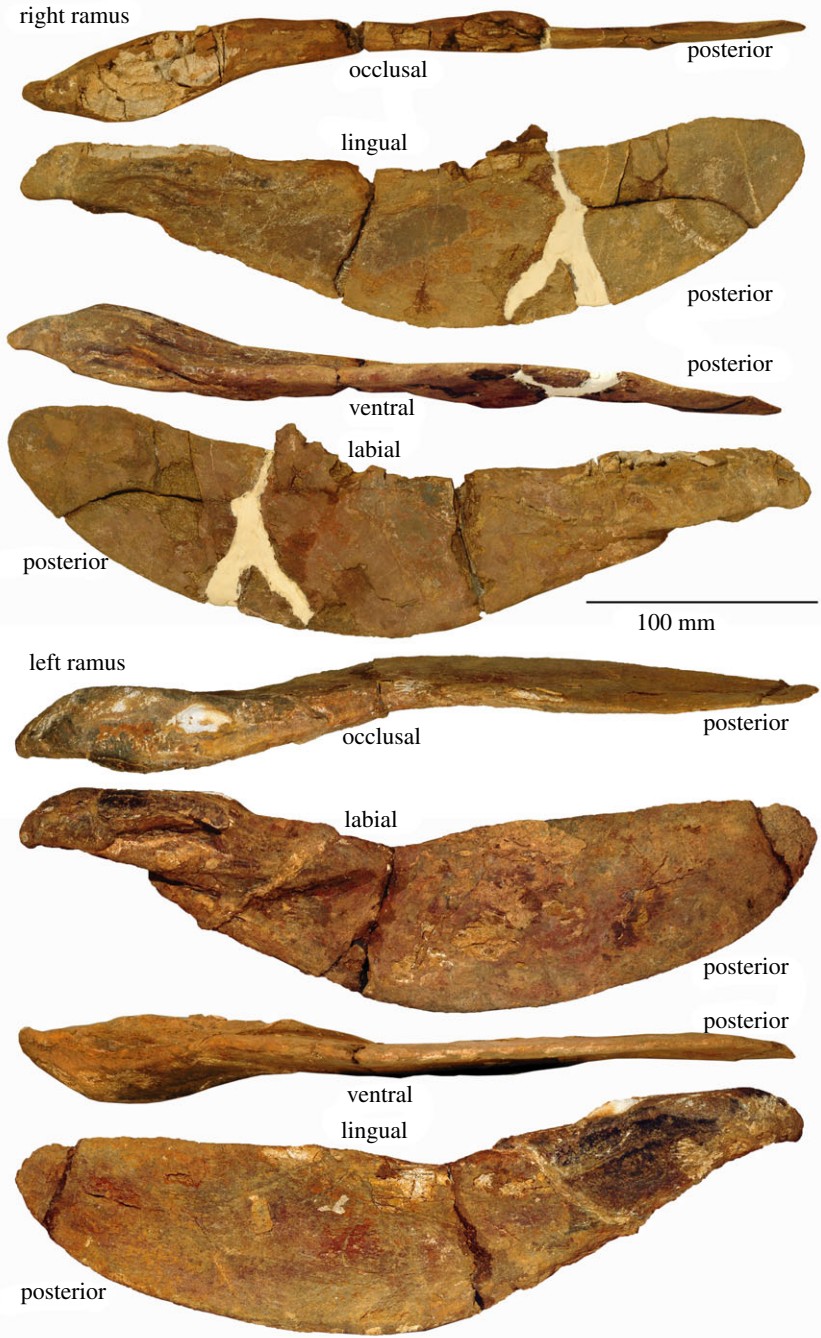

**Figure 2.** Inferognathals of *Ta. lavocati* (PIMUZ A/I 4717), from the Southern Maïder basin, Morocco. Photographed at the University of Zurich. Total length = 33 cm.

pattern common across all *Titanichthys* species with known gnathal, supragnathal or inferognathal elements [9].

   *Titanichthys* is considered to have been a member of the family Mylostomatidae, with *Bungartius perissus* and *Ta. lavocati* [8]—both of which are thought to have been durophagous, although there was little evidence of *Tafilalichthys* lower jaw elements prior to this paper [28]. Durophagy seems an extremely plausible feeding method for *Bungartius*, with a thickened occlusal surface at the anterior symphyseal region on its inferognathal appearing ideally suited to function as a shearing surface [37].

   To date, the only described *Tafilalichthys* jaw specimen is an anterior supragnathal [32], which indicated that *Tafilalichthys* was durophagous, although not specialized to the same degree as the related *Bungartius* or *Mylostoma* [28]. The *Tafilalichthys* inferognathal investigated herein (PIMUZ A/I 4717—figure 2) suggests that *Tafilalichthys* may have been more adapted for durophagy than

previously thought, with the anterior symphyseal region somewhat resembling that previously described for *Bungartius* and other durophagous arthrodires—with the occlusal dorsal surface partially composed of a cancellous texture [38]. However, this surface is flattened to the point of horizontality in *Tafilalichthys*, whereas both *Bungartius* [37] and *Mylostoma* [39] have more curved dental regions—which potentially could also have 'chopped' prey [40].

Like *Mylostoma*, the posterior 'blade' portion of *Tafilalichthys*' inferognathal comprises over half of the total length, as opposed to a smaller proportion in the earlier, Frasnian (383–372 Ma) mylostomatids—which were less specialized for durophagy [41]. This proportional lengthening of the blade is thought to have increased the area of attachment for the adductor (jaw-closing) muscles, thereby increasing the bite force; crucial when specializing upon tough-to-digest, hard-shelled prey [42].

*Dunkleosteus* was selected as a comparison due to its well-documented status as an apex predator [43] and an arthrodire—indicating fairly close relatedness with Mylostomatidae [8]. Ideally, a *Dunkleosteus marsaisi* specimen would have been the first preference to be selected for use, as it co-occurred with *Ti. termieri* in the Southern Maïder basin [44]; however, this did not prove possible. Instead, *Dunkleosteus terrelli*, known from the Cleveland Shale, was used. While *D. terrelli* was substantially larger than *D. marsaisi*, the skulls of the two species seem to have broadly similar shapes [9]. Given that all jaws in this study were scaled to the same length, using either species would be likely to yield broadly similar results.

The inferognathal of *D. terrelli* is more clearly differentiated into blade and dental portions than the other arthrodires in this study. The dental portion is divided into an anterior fang-shaped cusp, presumably for puncturing flesh, and a posterior sharp blade which occluded with a parallel bladed surface on the supragnathal [29]. This masticating, bladed surface is part of the dental portion of the inferognathal, separate from the edentulous posterior portion [36]. From a simple visual comparison, it appears much better-adapted for consuming large prey than *Titanichthys*.

## 2.2. Extant taxa

Sharks were selected as an extant comparison group due to the range of feeding strategies they display, including taxa with potentially analogous lifestyles to the three arthrodiran species investigated. The basking shark (*Ce. maximus*) is a megaplanktivore, approaching a body length of 12 m [45]. Being closely related to an apex predator it co-occurs with, the great white shark (*Ca. carcharias*), the basking shark seems analogous with the proposed ecological niche of *Titanichthys*. The whale shark (*Rhincodon typus*) would potentially have represented an even closer analogue for *Titanichthys*, having also evolved from durophagous ancestors [46], as seems likely for *Titanichthys*—unfortunately whale shark specimens could not be accessed for this study.

The great white shark is an ideal analogue for *Dunkleosteus*, being a lamniform shark (the same order as *Cetorhinus* [47]) with a powerful bite force befitting of an apex predator [48]. The horn shark *H. francisci* was selected for its durophagous lifestyle [49], making it analogous for the proposed feeding strategy of *Tafilalichthys*. However, it is not that closely related to the other sharks in this study; being in a different order, the Heterodontiformes [50]. Due to the absence of known durophagous species among lamniform sharks, *Heterodontus* is the most suitable candidate for a durophage related to *Cetorhinus*.

To provide a further comparison point, and potentially assess whether certain lower jaw structural changes were common among parallel evolutionary pathways, whales were also included in the analysis. The planktivorous blue whale (*B. musculus*) was compared with the killer whale (*O. orca*), an apex predator [51]. A third comparison species was not used because of the lack of durophagous whale species.

Due to the considerable evolutionary distance between the suspension-feeding mysticetes and macrophagous odontocetes—which diverged around 38 Ma [52]—this comparison may be somewhat less strong. When the investigated species are co-occurring sister taxa, like *Titanichthys* and *Tafilalichthys*, morphological differences are more likely to be driven by a single explanatory factor, such as divergence of function. There is a far greater possibility that differences between distantly related species are due to a myriad of different factors, the effects of which are hard to distinguish between. The results for the whales should be viewed with that caveat in mind.

## 2.3. Finite-element model construction

All jaw models were produced using surface scans of the original specimens. Some specimens had already been scanned prior to this research (table 1), those remaining were scanned at the University

**Table 1.** The specimens used in the study and the institutes in which they were scanned. Additional *Titanichthys* and *Tafilalichthys* specimens were observed at the University of Zurich to provide a more thorough insight into the species.

| specimen number | species | order | scanning institute |
|---|---|---|---|
| PIMUZ A/I 4716 | *Titanichthys termieri* | Arthrodira | University of Zurich |
| PIMUZ A/I 4717 | *Tafilalichthys lavocati* | Arthrodira | University of Zurich |
| CM6090 | *Dunkleosteus terrelli* | Arthrodira | Cleveland Museum of Natural History |
| BMNH 1978.6.22.1 | *Cetorhinus maximus* | Lamniformes | Natural History Museum, London |
| ZMA.PISC.108688 | *Heterodontus francisci* | Heterodontiformes | Zoological Museum, Amsterdam |
| ERB 0932 | *Carcharodon carcharias* | Lamniformes | ZNA hospital Antwerp |
| BMNH 1892.3.1.1 | *Balaenoptera musculus* | Mysticeti | Natural History Museum, London |
| NMML-1850 | *Orcinus orca* | Odontoceti | Idaho Museum of Natural History |

of Zurich using an Artec Eva light 3D scanner (Artec 3D). Surface scans were used instead of computerized tomography (CT) scans as the size and composition of some specimens rendered CT scanning extremely difficult. This, unfortunately, prevented the incorporation of internal features into the models; therefore, the jaws were treated as homogeneous structures. Doing so has previously yielded differing results to more accurate, heterogeneous models [48]. However, the surface scans should still prove valid for the purely shape-based comparison undertaken in this paper; although CT scanning would be essential for an assessment of the absolute performance of *Titanichthys'* jaw. While the shark jaws were originally CT scanned [53], only the surfaces were used to ensure methodological equivalence between species.

Jaw scans were processed, cleaned (removal of extraneous material and smoothing of fractures) and fused (where jaws were scanned in separate pieces) using a combination of Artec Studio 12 (Artec 3D), Avizo 9.4 (FEI Visualization Sciences Group) and MeshLab [54]. Jaw models were scaled to the same total length, as model size and forces applied had to be kept constant to ensure the analysis was solely investigating the effect of jaw shape on stress/strain resistance. Ideally, the models would have been scaled to the same surface area instead of length, as this typically produces stress comparisons of greater validity [55]. Similarly, scaling models to volume is most effective for comparing strain resistance. However, the extremely varied dentition among the various species skewed the results when models were scaled to either the same surface area or volume; an effect that has been noted previously [36]. Consequently, it was judged that equivocating model size using jaw length produced reasonably comparable models.

The muscle force applied to the jaws was adapted from a prior investigation of arthrodiran jaw mechanics [36], which primarily centred on a *D. terrelli* inferognathal. Consequently, all jaws were scaled to the length of the *D. terrelli* inferognathal scanned herein. The material properties proposed by Snively *et al.* [36], based on typical arthrodiran inferognathals, were applied to all jaw models. Treating each jaw as one homogeneous material, jaws were assigned a Young's modulus of 20 GPa and a Poisson's ratio of 0.3. A vertical force of 300 N was applied at the presumed central point of adductor mandibulae attachment. While this does not accurately represent the force exerted by the muscles, it is a decent approximation, given the absence of further skull material with which muscle action could be modelled [29]. Each jaw was constrained at the attachment point with the skull— typically on the dorsal surface at the posterior end of the articular bone. This constraint involved fixing a node at the attachment point for both translation and rotation in the *X*, *Y* and *Z* axes. Another constraint was applied to a node at the base of the anteriormost tooth (or the roughly analogous location proportionally for species with no discernible dentition), fixed for translation in the *Y*-axis—effectively simulating the dentition being suspended within an item of prey.

Each jaw scan was 'meshed'—divided into elements, comprising the three-dimensional volume of the jaw—in Hypermesh (Altair Hyperworks; Troy, MI, USA), whereupon forces, constraints and material properties were applied. Each loaded model was imported into Abaqus (Dassault Systèmes Simulia Corp., Providence, RI, USA), where FEA was performed. Every element comprises multiple nodes, which make up the outline of the element. Given the material properties of the model and the applied constraints, the deformation at each node can be simulated using FEA [31]. From these deformations, the stresses and strains experienced by each element within the model can be calculated.

The primary indicator selected was von Mises stress, which relates to the likelihood of ductile yielding causing a structure to fail [55]. Maximum principal stress distribution across the jaw was also analysed, as an indicator of the probability of brittle fracture. Given that bone responds in both ductile and brittle manners to stresses [56], recording both von Mises and maximum principal stress values should provide a more comprehensive profile of the jaws' robustness. The maximum principal strain value of each element was also recorded. The extent of strain experienced within a structure indicates the degree of deformation undergone by the structure; therefore, models with lower strain values are more resistant to deformation [55]. Experimentally, it was observed that proportional comparisons based on each of the three metrics produced extremely similar results (see electronic supplementary material). Consequently, only von Mises stress was used for further analysis, as it seemed to reflect structural robustness effectively.

It is important to emphasize that the values displayed herein are very unlikely to accurately represent the actual values of stress that the jaws would have experienced. Re-scaling of the jaw length, as well as assignment of equal material properties and applied forces, renders the absolute values irrelevant. Instead, these measures all served to validate comparisons between the different finite-element models. Consequently, it is the proportional differences between the stress values experienced across the respective jaws that should be the main focus of analysis, as the disparities observed will indicate the relative robustness of the jaw shapes.

## 2.4. Finite-element analysis

Initial comparison of stress distribution across the finite-element models will be purely visual, which has been used repeatedly to effectively distinguish mandibles by their dietary function [36,57]. This will enable qualitative assessment of the stress patterns in the respective jaws, highlighting regions of particularly high stress and enabling an approximation of the differing overall resistances to stress.

In order for quantitative comparison between the jaws to take place, the average von Mises stress values were recorded for each model, from every element across the model. Typically, mean values are used [58], but median values may prove more robust to being skewed by extreme values [59]. Consequently, both mean and median values were calculated for *Titanichthys*, whereupon the value of the respective metrics could be assessed. Averaging has the advantage of enabling comparison of total stress and strain resistance with a far greater degree of precision than from a purely visual comparison [60]. When combined with visual comparison, particularly weak or robust sections of the structure can still be identified. In addition, the Kruskal–Wallis tests were used to assess for significance in any disparities between species' median von Mises stress values [61]—although the massive sample sizes of the underlying data, with some models having over 200 000 elements, are likely to imbue even small differences with statistical significance.

However, averaging results can be skewed by element size, with smaller elements typically yielding more accurate results [62]. To combat this, an 'intervals method' has been proposed [60], which incorporates element volumes. This method could allow for considerably more effective comparison of finite-element models and, consequently, more precise distinction between feeding strategies.

## 2.5. Intervals method

The full method is described in the original paper [60], but will be outlined in brief here. Following FEA, all elements in the model are sorted by their von Mises stress value. These are then grouped into a number of 'intervals', each of which has an equal range of stress values. Fifty intervals proved the optimal amount in the original experiment, so are used in this test (however, as few as 15 intervals were still broadly effective at discriminating between dietary functions).

The cumulative volume of the elements represented in each of the 50 intervals can be calculated, then represented as a percentage of the total model volume. This represents the distribution of stresses across a model, characterizing the proportion of the elements experiencing particular stress levels. A principal component analysis (PCA) is performed, based on the percentage of jaw volume represented in each interval, plotting the jaw models on two axes (principal components) that should describe the majority of variation in stress distribution. Species with similar diets should group together to an extent, if the differences in stress distribution between feeding strategies can be categorized. This method has successfully distinguished between different dietary preferences in jaw models previously, although using species within the same genus [60], much more closely related than the species tested here. Experimentally, it was observed that the whale jaw models were poorly suited for direct

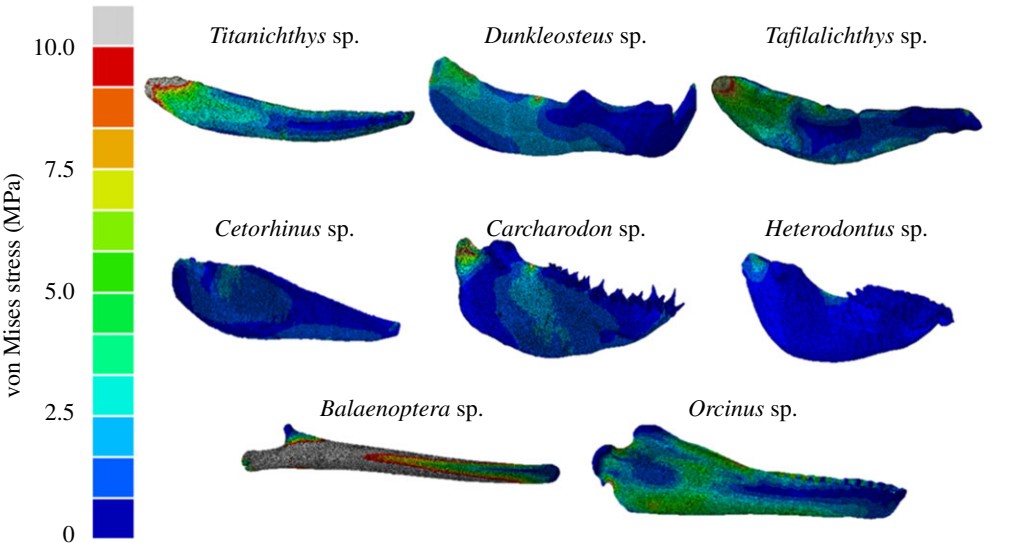

**Figure 3.** Von Mises stress distributions in the lower jaws of selected placoderm, shark and whale species, calculated using FEA (generally following the methodology of Snively *et al.* [36]).

comparison with the other species, as the considerable morphological disparity resulted in some models being represented in less than half of the stress intervals. Consequently, whales were removed from the PCA, to prevent skewing of the results.

# 3. Results

## 3.1. Visual comparison

The magnitudes of von Mises stress vary significantly between the placoderm inferognathals, but the general stress distribution patterns are relatively consistent (figure 3). The highest von Mises stress values for all three species occur in the posterior bladed region; particularly close to the jaw attachment point, probably as a result of the constraint applied there. Higher stress values are experienced on the lateral aspects of each jaw rather than on the medial. The fixed anterior point is also associated with high stress, but these regions are much more localized than at the jaw and muscle attachment points. *Titanichthys* exhibits the least resistance to von Mises stress among the placoderms, with *Dunkleosteus* proving the most resistant.

Visually, *Carcharodon* appears to be the least resistant to von Mises stress of the three shark lower jaws (figure 3), with *Heterodontus* probably the most resistant. In whales, the mandible of *Orcinus* is clearly more resistant than *Balaenoptera*, which is characterized by extremely high levels of von Mises stress, experienced across the majority of the structure (figure 3).

## 3.2. Quantitative comparison

Averaging the per element von Mises stress values produced differing results depending on whether the median or mean was used. However, while the actual values produced diverged (table 2), the proportional differences between the species remained relatively consistent. Consequently, either method seems equally applicable; to simplify the results, the median will henceforth be used as the method of averaging.

Among the placoderms, the inferognathal of *Titanichthys* was the least resistant by some margin (figure 4). The median elemental von Mises stress value for the inferognathal of *Tafilalichthys* represented 71% of the equivalent figure for *Titanichthys*, while in *Dunkleosteus*, it was just 37%.

In general, the average von Mises stress values for shark jaws (figure 4) were lower than in placoderms, with the highest value in sharks (in *Cetorhinus*) only slightly (0.1 MPa) higher than the lowest value in placoderms—for *Dunkleosteus*. While the jaws are typically more robust in sharks, there are some similar patterns when comparing proportional differences between the sharks. The suspension-feeding basking shark displays the highest average stress, although the difference between

**Table 2.** Average elemental stress and strain values for the lower jaws of various species of placoderms, sharks and whales, calculated using FEA. Both median and mean values are displayed. The unit for all values is MPa (megapascals).

| species | median | | | mean | | |
|---|---|---|---|---|---|---|
| | von Mises stress | maximum principal stress | maximum principal strain | von Mises stress | maximum principal stress | maximum principal strain |
| *Titanichthys termieri* | 1.83 | 0.65 | $4.95 \times 10^{-5}$ | 2.72 | 1.43 | $9.43 \times 10^{-5}$ |
| *Dunkleosteus terrelli* | 0.68 | 0.31 | $2.17 \times 10^{-5}$ | 0.94 | 0.51 | $3.29 \times 10^{-5}$ |
| *Tafilalichthys lavocati* | 1.29 | 0.48 | $3.81 \times 10^{-5}$ | 1.79 | 0.94 | $6.14 \times 10^{-5}$ |
| *Cetorhinus maximus* | 0.78 | 0.36 | $2.52 \times 10^{-5}$ | 0.88 | 0.51 | $3.18 \times 10^{-5}$ |
| *Carcharodon carcharias* | 0.58 | 0.26 | $2.01 \times 10^{-5}$ | 0.82 | 0.48 | $3.03 \times 10^{-5}$ |
| *Heterodontus maximus* | 0.22 | 0.11 | $7.78 \times 10^{-6}$ | 0.30 | 0.18 | $1.11 \times 10^{-5}$ |
| *Balaenoptera musculus* | 9.32 | 3.71 | $2.90 \times 10^{-4}$ | 11.66 | 6.56 | $4.17 \times 10^{-4}$ |
| *Orcinus orca* | 1.78 | 0.62 | $5.27 \times 10^{-5}$ | 2.14 | 1.22 | $7.65 \times 10^{-5}$ |

it and the macropredatory great white shark is notably smaller than the (potentially) corresponding disparity between *Titanichthys* and *Dunkleosteus*. The median von Mises stress for *Carcharodon* is 75% of the equivalent for *Cetorhinus*, a disparity dwarfed by the much greater resistance to stress observed in *Heterodontus* (28% of *Cetorhinus*).

With a median von Mises stress value of 9.32 MPa, the mandible of *B. musculus* is markedly less resistant to stress than all other jaws investigated (figure 4). There is a large inter-lineage disparity with the von Mises stress resistance of *Orcinus*, the median of which is 19% of that of *Balaenoptera*.

The Kruskal–Wallis tests revealed the differences between the median von Mises stress values for each species to be highly significant ($p < 0.0001$).

## 3.3. Intervals method

The intervals method PCA (figure 5) attempts to differentiate between species based on the distribution of stress across the jaw models. The method groups *Titanichthys* with the planktivorous *Cetorhinus* and the closely related *Tafilalichthys*. There is little obvious diet-based grouping of the macrophagous species, with the non-*Cetorhinus* shark species relatively close together.

# 4. Discussion

## 4.1. *Titanichthys*' jaw conforms to a suspension-feeding ecology

The inferognathal of *Titanichthys* was less resistant to von Mises stress than those of either *Dunkleosteus* or *Tafilalichthys*. *Dunkleosteus terrelli* has been substantively established as an apex predator [29,43], while *Tafilalichthys* can be considered to have been durophagous with some confidence, due to its morphological resemblance to, and close relatedness with, known durophagous arthrodires [8,28]. The comparatively high levels of stress observed in the inferognathal of *Titanichthys* suggest that neither feeding strategy would have been possible for *Titanichthys*, as its inferognathal would probably have failed (either by ductile yielding or brittle fracture) if exposed to the forces associated with the alternative feeding strategies. This strongly suggests that it was indeed a suspension feeder, as predicted based on its jaw morphology.

If *Titanichthys* were a suspension feeder, the primary function of its jaw would have been to maximize the water taken into the oral cavity during feeding, thereby increasing the rate of prey intake [63]. Morphologically, the inferognathal of *Titanichthys* seems ideally suited for this purpose—its elongation increased the maximum capacity of the oral cavity, which correlates with water filtration rate [64]. The perceived elongation of *Titanichthys*' inferognathal can be demonstrated by comparing its size with an inferognathal of the similarly sized *D. terrelli*, which is clearly wider and shorter—the specimen used here is less than half the length of the corresponding *Titanichthys* inferognathal [65]. The narrowing of

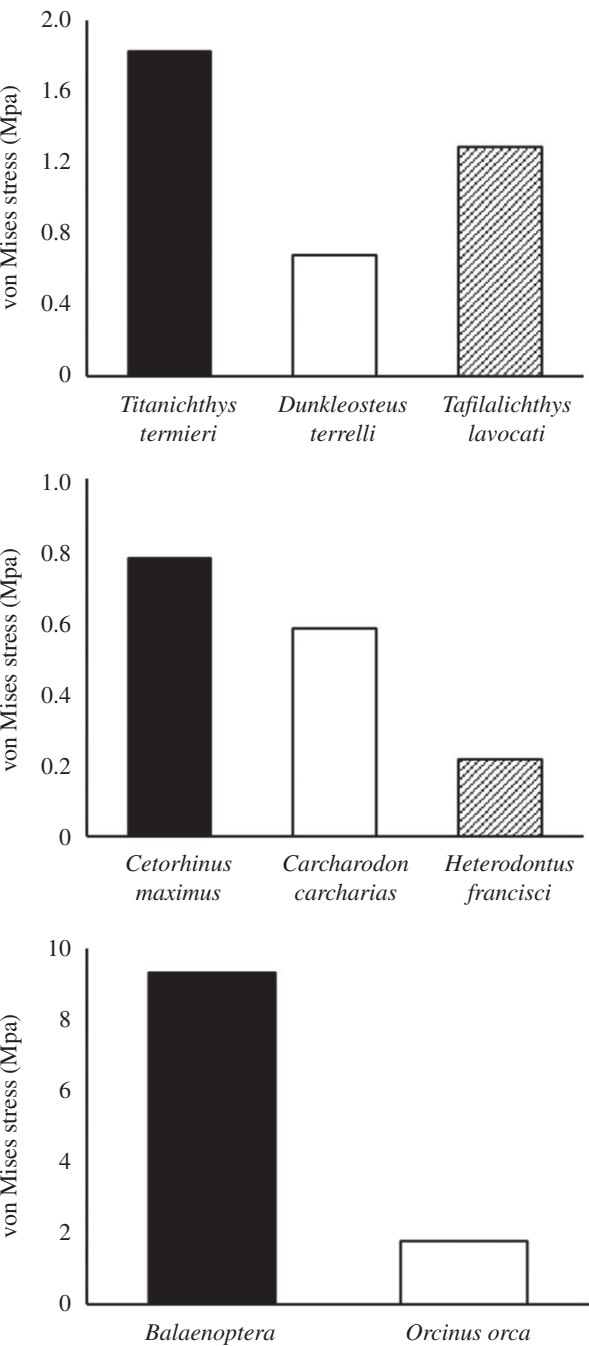

**Figure 4.** Median von Mises stress values for each jaw finite-element model. Bar colour corresponds with the potential ecological niche of each species.

*Titanichthys'* inferognathal, associated with elongation, would have reduced its mechanical robustness (as displayed in this study). This adaptation would probably be unfeasible for a species reliant upon consuming large or hard-shelled prey, as it would result in a fitness reduction from an adaptive peak [66].

While the inferognathals of both species were considerably more mechanically resilient than that of *Titanichthys*, there is still a sizable disparity between the von Mises stress values observed in *D. terrelli* and *Ta. lavocati*. Biomechanical analyses have suggested that *Dunkleosteus* was capable of feeding on both highly mobile and armoured prey [29], due to its high bite force and rapid jaw kinematics. In rodents, species with generalist diets have been shown to be more resistant to stresses across the skull than their more specialist relatives [67]. It is possible the comparatively generalist *Dunkleosteus* had a more stress-resistant inferognathal than the specialist durophage *Tafilalichthys* for the same reason.

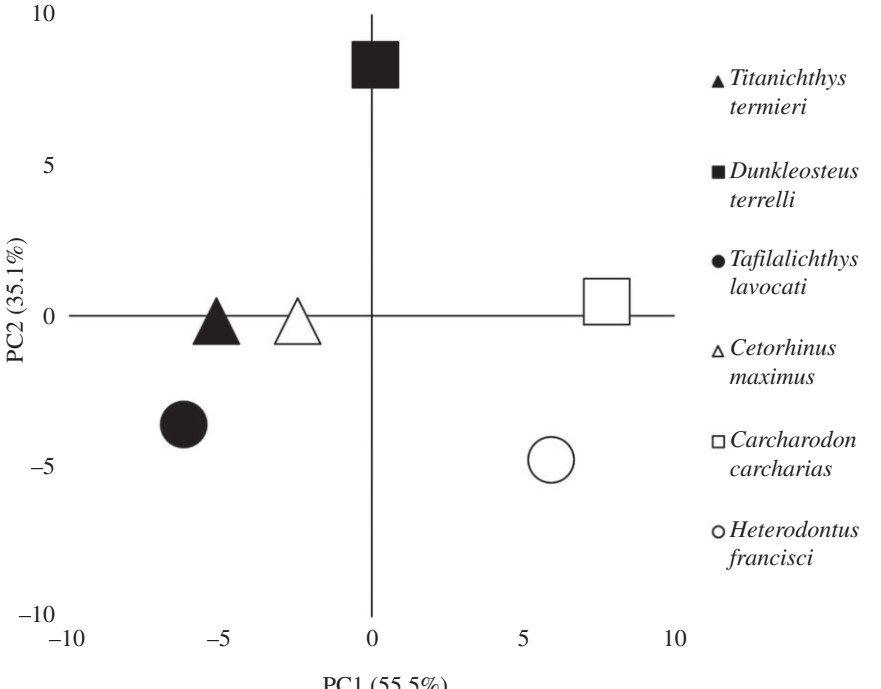

**Figure 5.** PCA visualizing the variation in von Mises stress distribution between the lower jaw finite-element models, as indicated by the intervals method [60]. Symbol colour is used to distinguish between clades: placoderm symbols are white and shark symbols are black. Shapes correspond with the potential ecological niche of each species. The percentage values on the axes indicate the variance explained by each principal coordinate. PC1 and PC2 cumulatively account for 90.6% of the total variance.

In sharks, the highest values of stress are seen in the suspension-feeding basking shark. This adds weight to the conclusion that *Titanichthys* was a suspension feeder, as the obligate planktivorous shark is significantly less resistant to stress than its durophagous and macropredatory relatives. The disparity in stress resistance between the lower jaws of *Carcharodon* and *Cetorhinus* is smaller than the equivalent disparity between *Titanichthys* and *Dunkleosteus*. The basking shark's lower jaw retains the same basic structure, albeit with less complexity, of the other shark species; whereas the lower jaw of *Titanichthys* is more morphologically divergent from the other placoderm species investigated, probably causing the more disparate results.

It is notable that, while there is a large difference in lower jaw robustness between *Cetorhinus* and *Titanichthys* (median von Mises stress of 0.78 MPa compared with 1.83 MPa, respectively), *Carcharodon* and *Dunkleosteus* performed very similarly. The median von Mises stress for *Carcharodon* was 85% of the respective value for *Dunkleosteus*. *Dunkleosteus* probably occupied the equivalent niche as *Carcharodon*, but their methods of subduing prey probably differed as a result of very efficient locomotion in the great white shark [68], which is unlikely to have been replicated in the heavy, less streamlined *Dunkleosteus* [69]. Similarly, some predatory strategies of *Carcharodon*, like the lateral head shake [70], may not have been plausible for *Dunkleosteus*. Consequently, the great white shark lower jaw was expected to prove more resistant to stress than the inferognathal of *Dunkleosteus*—and this may have been seen to a greater extent if cartilaginous properties were applied to the shark. Treating a great white shark jaw as homogeneous bone has previously resulted in underestimated stress resistance [48], and a lower Young's modulus associated with calcified cartilage would result in higher jaw strain. On the other hand, prior research indicating that the bite force : body mass ratio of *Dunkleosteus* is roughly equivalent to that of the great white shark [29] suggests that similar stress resistances, when scaled to length, are to be expected.

The stress resistance of the great white shark's lower jaw may have been roughly equivalent to that of *Dunkleosteus*, but no such resemblance between potential analogues was observed in the durophagous species. The lower jaw of *H. francisci* is a thick structure devoid of ornamentation beyond its dentition, which proved to be substantially more robust than the lower jaw of any other species investigated. The mass-specific bite force of *H. francisci* has been shown to markedly exceed that of *Ca. carcharias* [71], enabling efficient crushing of its hard-shelled prey. Consequently, the disparity in stress resistance between the two species is not unexpected.

What initially seems more surprising is the even larger difference between the jaw robustness of the two durophagous species, with the horn shark being far more resilient than *Tafilalichthys*. Their roughly equivalent diets would suggest similar mechanical requirements of their jaws; however, the disparity may be explained by behavioural differences. Durophagous placoderms are thought to have primarily broken down the hard shells of their prey using shearing, as opposed to the more mechanically taxing, crushing mechanism seen in chondrichthyans and other post-Devonian fish [72]. This suggests that the jaws of *Tafilalichthys* would have experienced less stress than those of the shell-crushing *Heterodontus* [49].

It is worth noting that the shark finite-element models were produced using surface scans originally created for use in a geometric morphometric study [53]. Consequently, they were not ideally suited to being discretized into a single, uniform surface. Despite extensive remeshing using both Blender [73] and Hypermesh, the shark jaw models were still of poorer quality than the other jaw models. The impact of this on the overall results is difficult to determine, but it should be kept in mind that the broad patterns are of more utility and interest than any specific numerical values.

The fundamental pattern outlined within this study is demonstrated further in whales: the mandible of the suspension-feeding blue whale is less resistant to von Mises stress than that of the macropredatory killer whale, but with a far greater disparity than in the other lineages. Jaw elongation is seen to a far greater degree in the mysticete whales than the other megaplanktivorous lineages investigated, probably as a consequence of the energetically expensive 'lunge feeding' method used by most mysticetes [74]. This is doubly true for the massive jaws of the blue whale, which enable incredibly efficient feeding despite substantial mechanical expenditure [75]. Consequently, resistance to stress may be lowest in the blue whale jaw as a result of maximizing feeding efficiency.

The mandible of *Orcinus* is considerably more resistant to stress than that of *Balaenoptera*, but the median values are still notably higher than in *Carcharodon* and *Dunkleosteus*, the proposed analogues of the killer whale. Ecological reasons for this are difficult to determine, with the typical diet of an orca resembling the diet of a great white shark: centring on marine mammals [76] but sufficiently generalist to predate a wide range of species [77]. This ecological similarity would seem to suggest roughly equivalent jaw robustness, a pattern, which is not seen here.

Methodological factors may have impacted the modelling results for the whale jaws. The orca's teeth were not attached to the scanned mandible. Teeth were generally associated with relatively low stress values in this study, removing these regions from the model may have raised the average values. Manually attaching them to the digital model was considered, but the imperfect nature of this would probably have further reduced the validity of the model; similarly, removing the dentition from basking shark jaws or the bone parts used for cutting or crushing in placoderm jaws would have been impossible.

All jaw scans were scaled to the same length, to circumvent the impact of teeth on scaling to the same surface area. While this seemed to improve the validity of comparisons between the model placoderm and shark jaws, it may have had the opposite effect with the whale jaws. Scaling the blue whale jaw rendered it extremely narrow relative to the other jaws, to an unrealistic extent. This may partially explain the average stress value calculated in the blue whale jaw massively exceeding those of any other species. Indeed, when the whale jaws were scaled to the same surface area, the average stress values in the orca's jaw were around 70% of the equivalent values in the blue whale. By contrast, re-scaling had little impact on the orca's jaw robustness compared with the other apex predators. Again, the large-scale trends are much more valuable than any specific numerical values—and using either method revealed that the megaplanktivore jaw was significantly less mechanically resilient than that of the apex predator.

The intervals method is probably better-suited to comparing between more closely related species [60], as their morphology would probably be more homogeneous—making function-related divergences more central to the analysis. Despite the vast evolutionary distances involved, the method effectively grouped the planktivorous *Cetorhinus* together with *Titanichthys*. This may suggest that *Titanichthys* was a suspension feeder, as the jaws of *Titanichthys* and *Cetorhinus* are very distinct morphologically, yet consistent patterns in von Mises stress distribution between them are statistically quantifiable. The close placement of *Titanichthys* and *Tafilalichthys* does suggest some caution should be taken with any interpretation, although this is probably more a function of their close relatedness than of a shared ecological niche. The PCA did not group the durophagous or macropredatory species together, although this was predictable to an extent as the stress values of those species seemed to be more influenced by their lineage than their diet. Despite this, the intervals method's detection of corresponding stress patterns between (potentially) suspension-feeding species is notable. This method should be applied in a variety of contexts moving forward, to assess for mechanical adaptations underlying functional divergences in other lineages.

## 4.2. Trajectories in megaplanktivory: durophagous origins?

Complete *Tafilalichthys* inferognathals have not previously been figured in the literature. Consequently, the specimen described in this study is significant for advancing our understanding of arthrodiran interrelationships and the evolutionary pathway that seemingly resulted in obligate planktivory in *Titanichthys*. The morphology and mechanical performance of the inferognathal both indicate that durophagy was the most likely feeding strategy for *Tafilalichthys*, supporting its proposed phylogenetic position within the Mylostomatidae [8]. With all the major mylostomatids (excluding *Titanichthys*) likely to have been durophagous, it seems reasonable to conclude that *Titanichthys* evolved from durophagous ancestors.

Evolutionary transitions from durophagy to planktivory have occurred a number of times. The suspension-feeding whale shark (*Rhincodon typus*) arose from the typically benthic Orectolobiformes [46]. Its closest relative, the nurse shark (*Ginglymostoma cirratum*), is durophagous, feeding principally on hard-shelled invertebrates [78]. Similarly, the sister taxon of the planktivorous Mobulidae (manta and devil rays) are the durophagous Rhinopteridae (cownose rays) [79,80]. In a less clear parallel, the only pinniped proposed to have been durophagous [81] was relatively closely related to the ancestor of the Lobodontini—pinnipeds uniquely specialized for planktivory [82].

## 4.3. Temporal evolution and extinction of megasuspension feeders

The emergence of a megaplanktivore in the Famennian may hold similar clues to the degree and nature of marine primary productivity during the Devonian period [83]. Modern forms migrate to regions of high seasonal productivity, such as mysticetes seeking arctic oceans and highly productive upwelling zones. Basking sharks focus on relatively less productive seasonal blooms in shallow boreal and warm temperate waters, while whale sharks are associated with tropical waters and seasonal blooms and spawning events in this realm. The evolution of megaplanktivores coincided with periods of high productivity [16,17]. For example, the radiation of mysticete whales coincides with the Neogene cooling pump and the onset of the circumantarctic polar current, resulting in a stronger thermohaline pump. It has been noted that the emergence of suspension-feeding pachycormid fishes correlates with the evolution of key phytoplankton: dinoflagellates, diatoms and coccolithosphorids could reflect the increase in primary productivity that led to the Mesozoic marine revolution [84]. While perhaps not necessarily being drivers of the revolution, the conditions permissive of such a radiation in marine primary producers may indeed reflect a marked shift in opportunity. Similarly, suspension-feeding radiodonts during the Cambrian explosion [4,85,86] radiated synchronously with the first establishment of a tiered food chain with several (at least four) levels of consumers. While the Cambrian radiation may be entirely unique with the innovation of micropredation [87], evidence for increased primary productivity is manifested in global Early Cambrian phosphate deposits [88], often associated with upwelling systems in modern oceans [89].

The Devonian saw the first emergence of arborescent plants on land [24]. This resulted in deeper rooting systems, higher silicate rock weathering and nutrient run off into the oceans. While increasing primary productivity, it also led to near global deep ocean anoxia, black shale deposition [90,91] and the Frasnian–Famennian Kellwasser event, one of the 'big five' mass extinctions [92]. The increased nutrients going into circulation may well have been the necessary push for allowing arthrodires to explore this ecological niche of megaplanktivory as the first vertebrates on record.

The apparent punctuation and compelling correlation between major marine radiations, shifts in apparent productivity and megaplanktivores may be of interest for understanding how this unique ecological strategy responds to global perturbations, such as human-induced climate change. With their potential added sensitivity, megaplanktivores may be 'canary birds' for ocean ecosystem health. Some caution is advised, however. There may be taphonomic biases preventing the recognition of each and every megaplanktivore in existence at a given time. As with *Titanichthys*, tell-tale features of ecology may have been lost during fossilization. As a rule, one would want to have the suspension-feeding apparatus preserved, but otherwise other associated anatomical adaptations or stomach contents will need to be identified [93].

There are almost certainly other planktivorous species in the fossil record yet to be identified, shown by the recent re-appraisal of the Cretaceous plesiosaur *Morturneria seymourensis* as a probable suspension feeder [94]. Indeed, there are even other placoderms that may have been planktivorous: the arthrodire *Homostius* had a narrow jaw devoid of dentition or shearing surfaces and substantially pre-dated *Titanichthys* [95]. The common reduction in stress/strain resistance observed here could be used as an indicator of planktivory in such cases where it seems plausible but cannot be identified definitively, due to the absence of fossilized suspension-feeding structures.

# 5. Conclusion

FEA of the lower jaw of *Titanichthys* revealed that it was significantly less resistant to von Mises stress than those of related arthrodires that used macrophagous feeding strategies. This suggests that these strategies would not have been viable for *Titanichthys*, as its jaw would have been insufficiently mechanically robust. Consequently, it is highly likely that *Titanichthys* was a suspension feeder—a feeding method that is likely to exert considerably less stress on the jaw than macrophagous feeding modes. The validity of assigning suspension feeding based on jaw mechanical resilience is supported by the roughly equivalent patterns known from lineages containing extant suspension feeders.

Common morphological trends in the convergent evolution of megaplanktivores can not only be observed but quantified mechanically using FEA. A variety of methods were used to compare between the jaw models, due to imperfections with solely comparing visually or using average stress. The intervals method grouped feeding strategies to an extent, providing an additional perspective.

*Tafilalichthys*, probably a member of the Mylostomatidae and, therefore, one of *Titanichthys*' closest relatives, appears to have been durophagous. Durophagy is the likely feeding mode of all crown-group mylostomatids except *Titanichthys*, suggesting that it evolved from a durophagous ancestor. This durophage-to-planktivore transition is surprisingly common among convergently evolved giant suspension feeders: it is also seen in multiple, independently evolved planktivorous elasmobranch lineages.

The presence of a megaplanktivore in the Famennian supports the theory that productivity was high in the Late Devonian, which was probably a result of increased eutrophication caused by the diversification of terrestrial tracheophytes and the advent of arborescence. It reflects the link between the increasing complexity of Devonian marine ecosystems and the functional diversity of Arthrodira, which occupied a wide range of ecological niches. Most significantly, it reveals that vertebrate megaplanktivores probably existed over 150 Ma prior to the Mesozoic pachycormids, previously considered the earliest definitive giant suspension feeders.

Data accessibility. All data available from the Dryad Digital Repository: https://doi.org/10.5061/dryad.9kd51c5d6 [96]. Jaw scans used with the permission of museums have not been made available in their raw format, but the finite-element models produced using them are accessible.

Authors' contributions. S.J.C., E.J.R. and J.V. designed the study. C.K. provided and scanned the Moroccan placoderm specimens. S.J.C. carried out all analysis and wrote the manuscript, with critical revision from all authors. All the authors gave final approval for submission.

Competing interests. We declare that we have no competing interests.

Funding. C.K. was supported by the Swiss National Science Foundation SNF.

Acknowledgements. We would like to thank Jordi Marcé-Nogué, for allowing us to use his technique and adjusting it for our data. Jaw scans were provided by the Natural History Museum in London, the Idaho Museum of Natural History and Matt Friedman; others were acquired from Pepijn Kamminga's online repository of shark jaw models [49]—we extend our gratitude to them all. We also thank James Boyle for his insight and translation of relevant literature. We would like to thank two anonymous reviewers for their constructive feedback.

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
