## [Reviewer comments · Royal Society Open Science]

Review History

RSOS-200272.R0 (Original submission)

Review form: Reviewer 1 (Jeff Liston)

Is the manuscript scientifically sound in its present form?

Yes

Are the interpretations and conclusions justified by the results?

Yes

Is the language acceptable?

Yes

Do you have any ethical concerns with this paper?

No

Have you any concerns about statistical analyses in this paper?

No

Recommendation?

Accept with minor revision (please list in comments)

Comments to the Author(s)

This is a solid piece of work and a great resolution to a question which has vexed me for some time, namely the assessment of an animal as suspension-feeder, in the absence of any preserved structures that can be conclusively identified as functionally suspension-feeding (see also Liston 2008, 41-49 for suspension-feeding structures that rarely preserve at all, yet are clearly critical to suspension-feeding) - a question which has been particularly relevant for the debate over the trophic role of *Titanichthys*. However, the very nature of the objects in question, in terms of their large size and sometimes fragility, introduces its own problems, which have understandably prevented the authors from treating all specimens equitably. In particular, as noted by the authors, CT scans would be desirable for all objects for examination of the internal structure to correlate density variations throughout each ramus-structure (particularly as a load borne hanging open while it moves forward). In this regard, analysis of *Leedsichthys* lower jaws would be particularly enlightening with its advanced bone resorption, but understandably this has been omitted, and perhaps would be a worthwhile follow-on project to this work. In some ways, this work raises almost as many questions as it answers, but it is clearly a very welcome leap forward. For terminology, suspension-feeding and filter feeding are used interchangeably throughout the text when suspension-feeding should be exclusively employed until the mechanism of extracting the particles is clarified (see in particular Liston 2007 p.214-232 for summary of the literature on this), as it implies use of the structure as a sieve or filter when the mechanisms and their physical uses seem to be far more complex than that for contemporary - never mind extinct - fish. All that said, the annotated file (Appendix A) is mostly tiny typos and suggested rephrasings, and I applaud the authors for grasping this nettle, and hope that this work continues further.

Review form: Reviewer 2**Is the manuscript scientifically sound in its present form?**

Yes

Are the interpretations and conclusions justified by the results?

Yes

Is the language acceptable?

Yes

Do you have any ethical concerns with this paper?

No

Have you any concerns about statistical analyses in this paper?

No

Recommendation?

Accept with minor revision (please list in comments)

Comments to the Author(s)

This is a great paper.

I have made a few comments onto the manuscript (Appendix B), but these are small observations.

Decision letter (RSOS-200272.R0)

06-Apr-2020

Dear Mr Coatham,

On behalf of the Editors, I am pleased to inform you that your Manuscript RSOS-200272 entitled "Was the Devonian Placoderm *Titanichthys* a Filter-Feeder?" has been accepted for publication in Royal Society Open Science subject to minor revision in accordance with the referee suggestions. Please find the referees' comments at the end of this email.

The reviewers and handling editors have recommended publication, but also suggest some minor revisions to your manuscript. Therefore, I invite you to respond to the comments and revise your manuscript.

- Ethics statement

- Data accessibility

<http://datadryad.org/submit?journalID=RSOS&manu=RSOS-200272>

- Competing interests

- Authors' contributions

AB carried out the molecular lab work, participated in data analysis, carried out sequence alignments, participated in the design of the study and drafted the manuscript; CD carried out the statistical analyses; EF collected field data; GH conceived of the study, designed the study,

coordinated the study and helped draft the manuscript. All authors gave final approval for publication.

- Acknowledgements

- Funding statement

Because the schedule for publication is very tight, it is a condition of publication that you submit the revised version of your manuscript before 15-Apr-2020. Please note that the revision deadline will expire at 00.00am on this date. If you do not think you will be able to meet this date please let me know immediately.

If your manuscript is newly submitted and subsequently accepted for publication, you will be asked to pay the article processing charge, unless you request a waiver and this is approved by Royal Society Publishing. You can find out more about the charges at <https://royalsocietypublishing.org/rsos/charges>. Should you have any queries, please contact opscience@royalsociety.org.

Kind regards,
Andrew Dunn
Royal Society Open Science Editorial Office
Royal Society Open Science
opscience@royalsociety.org

on behalf of Professor Marcelo Sanchez (Associate Editor)
opscience@royalsociety.org

Reviewer comments to Author:

Reviewer: 1
Comments to the Author(s)

This is a solid piece of work and a great resolution to a question which has vexed me for some time, namely the assessment of an animal as suspension-feeder, in the absence of any preserved structures that can be conclusively identified as functionally suspension-feeding (see also Liston 2008, 41-49 for suspension-feeding structures that rarely preserve at all, yet are clearly critical to suspension-feeding) - a question which has been particularly relevant for the debate over the trophic role of *Titanichthys*. However, the very nature of the objects in question, in terms of their large size and sometimes fragility, introduces its own problems, which have understandably prevented the authors from treating all specimens equitably. In particular, as noted by the authors, CT scans would be desirable for all objects for examination of the internal structure to correlate density variations throughout each ramus-structure (particularly as a load borne hanging open while it moves forward). In this regard, analysis of *Leedsichthys* lower jaws would be particularly enlightening with its advanced bone resorption, but understandably this has been omitted, and perhaps would be a worthwhile follow-on project to this work. In some ways, this work raises almost as many questions as it answers, but it is clearly a very welcome leap forward.

For terminology, suspension-feeding and filter feeding are used interchangeably throughout the text when suspension-feeding should be exclusively employed until the mechanism of extracting

the particles is clarified (see in particular Liston 2007 p.214-232 for summary of the literature on this), as it implies use of the structure as a sieve or filter when the mechanisms and their physical uses seem to be far more complex than that for contemporary - never mind extinct - fish.

All that said, the annotated file is mostly tiny typos and suggested rephrasings, and I applaud the authors for grasping this nettle, and hope that this work continues further.

Reviewer: 2

Comments to the Author(s)

This is a great paper.

I have made a few comments onto the manuscript, but these are small observations.

Author's Response to Decision Letter for (RSOS-200272.R0)

See Appendix C.

Decision letter (RSOS-200272.R1)

23-Apr-2020

Dear Mr Coatham,

It is a pleasure to accept your manuscript entitled "Was the Devonian Placoderm *Titanichthys* a Suspension-Feeder?" in its current form for publication in Royal Society Open Science.

Best regards,
Lianne Parkhouse

Editorial Coordinator
Royal Society Open Science
openscience@royalsociety.org

on behalf of Professor Marcelo Sanchez (Associate Editor) and the Subject Editor.
openscience@royalsociety.org

Follow Royal Society Publishing on Twitter: [@RSocPublishing](https://twitter.com/RSocPublishing)

Appendix A**ROYAL SOCIETY
OPEN SCIENCE****Was the Devonian Placoderm *Titanichthys* a Filter-Feeder?**

Journal:	Royal Society Open Science
Manuscript ID	RSOS-200272
Article Type:	Research
Date Submitted by the Author:	20-Feb-2020
Complete List of Authors:	Coatham, Sam; The University of Manchester, Earth Sciences Vinther, Jakob; University of Bristol, Biological Sciences Rayfield, Emily; University of Bristol, School of Earth Sciences Klug, Christian; Universität Zürich, Paläontologisches Institut und Museum
Subject:	Palaeontology < EARTH SCIENCES, biomechanics < BIOLOGY, evolution < BIOLOGY
Keywords:	Filter-Feeding, Titanichthys, Arthrodira, Finite Element Analysis, Placodermi
Subject Category:	Earth and Environmental Science

Author-supplied statements

Relevant information will appear here if provided.

Ethics

Does your article include research that required ethical approval or permits?:

This article does not present research with ethical considerations

Statement (if applicable):

CUST_IF_YES_ETHICS :No data available.

Data

It is a condition of publication that data, code and materials supporting your paper are made publicly available. Does your paper present new data?:

Yes

Statement (if applicable):

All data has been made accessible at the Dryad Digital Repository (<https://doi.org/10.5061/dryad.9kd51c5d6>, review URL: https://datadryad.org/stash/share/sA4rCcK0slzvPgaU3QAzH7wwIlg8_Q2JI9mLPpsCQjgg). Jaw scans used with the permission of museums have not been made available in their raw format, but the finite element models produced using them are accessible.

Conflict of interest

I/We declare we have no competing interests

Statement (if applicable):

CUST_STATE_CONFLICT :No data available.

Authors' contributions

This paper has multiple authors and our individual contributions were as below

Statement (if applicable):

S.J.C, E.J.R and J.V designed the study. C.K provided and scanned the Moroccan placoderm specimens. S.J.C carried out all analysis and wrote the manuscript, with critical revision from all authors. All the authors gave final approval for submission.

Was the Devonian Placoderm *Titanichthys* a Filter-Feeder?

Samuel J. Coatham^{1*†}, Jakob Vinther¹, Emily J. Rayfield¹ and Christian Klug²

¹ Life Sciences Building, University of Bristol, 24 Tyndall Avenue, Bristol; e-mails:

sam.coatham@postgrad.manchester.ac.uk, jakob.vinther@bristol.ac.uk, E.rayfield@bristol.ac.uk

² Paläontologisches Institut und Museum, Universität Zürich, Karl-Schmid-Strasse 4, 8006 Zürich; e-mail: chklug@pim.uzh.ch

Keywords: Filter-feeding, *Titanichthys*, Arthrodira, Devonian, comparative biomechanics

1. Summary

Large nektonic suspension feeders have evolved multiple times. The apparent trend among apex predators for some evolving into feeding on small zooplankton is of interest for understanding the associated shifts in anatomy and behaviour while the spatial and temporal distribution gives clues to an inherent relationship with ocean primary productivity and how past and future perturbations to these may impact on the different tiers of the food chain. The evolution of large nektonic suspension feeders - 'gentle giants' - occurred 4 times among chondrichthyan fishes (e.g. whale and basking sharks and manta rays) and baleen whales (mysticetes), the Mesozoic pachycormid fishes and at least twice in radiodontan stem group arthropods (Anomalocaridids) during the Cambrian Explosion. The Late Devonian placoderm *Titanichthys* has tentatively been considered to have been a megaplanktivore, primarily due to its gigantic size and narrow, edentulous jaw while no filtering apparatus have ever been reported. Here the potential for microphagy and other feeding behaviours in *Titanichthys* is assessed via a comparative study of jaw mechanics in *Titanichthys* and other placoderms with presumably differing feeding habits (macrophagy and durophagy). Finite element models of the lower jaws of *Titanichthys termieri* in comparison to *Dunkleosteus terrelli* and *Tafilalichthys lavocati* reveal considerably less resistance to von Mises stress in this taxon. Comparisons with a selection of large-bodied extant taxa of similar ecological diversity reveals similar disparities in jaw stress resistance. Our results therefore conform to the hypothesis that *Titanichthys* was a filter-feeder with jaws ill-suited for biting and crushing but well suited for gaping ram feeding.

2. Introduction

Some of the largest organisms ever to have roamed the ocean and alive today are suspension feeders. The switch to feeding on the lowest levels of the trophic pyramid is a tremendous shift in food resource [1]. While pursuing large bodied prey results in adaptations towards stealth, complex hunting behaviours and expanded sensory repertoires, suspension feeding results in a host of anatomical, migratory and behavioural modifications. Locomotory speed and energy reserves scale with body mass - enabling a migratory lifestyle in some species [2] to capitalise on seasonal periods of high food abundance [3]. Invertebrate filter-feeders are known from the Cambrian [4], giant-bodied relative to their temporal counterparts. While the first definitive vertebrate megaplanktivores occurred in the Mesozoic, within the pachycormids [5], this ecological niche may in fact have originated in the Devonian.

The arthrodire *Titanichthys* occurred in the Famennian [6], the uppermost stage of the Devonian (372-359 Ma [7]). There are multiple morphological features indicating that *Titanichthys* may have been a megaplanktivore, primarily its massive size [8]. The elongate, narrow jaws lack any form of dentition or shearing surface [9]; seemingly ill-equipped for any form of prey consumption more demanding than simply funnelling prey-laden water into the oral cavity. *Titanichthys* is also known for its small orbitals (relative to its size), indicating that visual acuity may not have been that important in its predatory behaviour. This is a known feature of predation in extant filter-feeders [10], so may be further evidence of planktivory. However, the filter-feeding

*Author for correspondence (sam.coatham@postgrad.manchester.ac.uk).

†Present address: Department of Earth and Environmental Sciences, Michael Smith Building, University of Manchester, Dover Street.

1 pachycormid *Rhinconichthys* has enlarged sclerotic rings [11], bringing into question the use of reduced
2 orbitals as a diagnostic character of planktivory.

3
4 Despite the numerous physical traits shared between *Titanichthys* and other definitive giant filter-feeders,
5 planktivory in *Titanichthys* has yet to be strongly supported, due to the absence of evidence of a filtering
6 structure. If *Titanichthys* was indeed a filter-feeder, presumably it would have fed in a roughly analogous
7 manner to modern planktivorous fish, ~~which strain prey from water exiting the oral cavity through the gills~~
8 ~~using elaborate gill rakers~~ [11] was also the filtering method of planktivorous pachycormids [11]. Placoderm
9 gill arches are rarely preserved [13], so the absence of a fossil filtering structure may be an artefact of the poor
10 fossil record, or it may indicate that *Titanichthys* was not a filter-feeder.

11
12 The viability of filter-feeding in *Titanichthys* is promoted by seemingly favourable conditions in the Devonian.
13 Increases in primary productivity appear to be associated with the recurrent evolution of megaplanktivores,
14 with potential expansions of available food resources enabling larger body sizes. This has been observed in the
15 diversification of mysticetes [14,15] and the origin of most filter-feeding elasmobranch clades [16], with
16 potential further correlations in the evolution of giant planktivorous anomalocarids in the Lower Cambrian
17 [17] and pachycormids in the Jurassic [18]. Productivity probably also increased throughout the Devonian,
18 with the combination of tracheophyte proliferation [19] and the advent of arborescence [20] likely accelerating
19 the rate of chemical weathering [21]. This could have resulted in enrichment of the oceanic nutrient supply via
20 runoff [22], potentially increasing marine productivity [18]. Although there is little direct proof of this [23], the
21 rise in diversity of predators with high energetic demands [24] indicates sufficient productivity to support
22 relatively complex ecosystems. Consequently, it seems probable that productivity did increase, potentially
23 facilitating the evolution of a giant filter-feeder in the Devonian.

24
25 To assess whether *Titanichthys* was indeed a filter-feeder, we investigated the mechanical properties of its jaw
26 in order to infer function. The engineering technique finite element analysis (FEA) has previously been used to
27 effectively differentiate between the mandibles of related species with differing diets [25]. Consequently, finite
28 element models of the inferognathals of *Titanichthys termieri*, *Tafilalichthys lavocati* and *Dunkleosteus terrelli* were
29 generated and compared. *Tafilalichthys* is thought to have been durophagous (specialised to consume hard-
30 shelled prey) [26], while *Dunkleosteus* was almost certainly an apex predator [27]; representing the two most
31 plausible feeding modes for *Titanichthys* (excluding planktivory). Both species were arthrodires related to
32 *Titanichthys*, with *Tafilalichthys* more closely related – likely within the same family [8].

33
34 By digitally discretising a structure into many elements and applying loads, constraints and material
35 properties, the stress and strain experienced within each element can be calculated in FEA [28]. When viewed
36 as components of the entire structure, its resistance to stress and strain can be clearly visualised, enabling
37 functional inference. While the magnitude of stress/strain values in extinct taxa are hard to definitively
38 ascertain, comparing between models loaded in the same manner is effective for comparative studies of
39 function [29]. Therefore, the mechanics of the arthrodire inferognathals will be compared based purely on
40 their shape. Extant taxa, the lifestyles of which are far better understood, will be used as a further reference
41 point, to validate the use of jaw robustness as a proxy for feeding strategy. The sharks *Cetorhinus maximus*
42 (basking), *Carcharodon carcharias* (great white) and *Heterodontus francisci* (horn) all occupy ecological niches
43 roughly analogous to those of the placoderms studied (planktivore, apex predator and durophage,
44 respectively). In addition, the cetaceans *Balaenoptera musculus* (blue whale) and *Orcinus orca* (killer whale) will
45 serve as a further planktivore-apex predator reference; albeit with much greater evolutionary distance
46 between the species.

47
48 Comparing the jaw mechanics of definitive filter-feeders with their macrophagous relatives will provide
49 clarity regarding the implications of any differences in stress/strain patterns of the placoderm jaws, informing
50 any conclusions regarding *Titanichthys*' feeding strategy. Should *Titanichthys* have been a filter-feeder, its jaw
51 would be expected to be less mechanically robust than those of related species with diets associated with
52 greater bite forces, which would exert more stress on the jaw. Consequently, the jaw of a filter-feeder is
53 predicted to be less resistant to stress and strain than those of the compared durophagous and
54 macropredatory species.

3. Materials and Methods

Placoderm Specimens

Titanichthys specimens are mostly known from the Cleveland Shale, with remains of five different *Titanichthys* species having been found there – albeit mostly from relatively incomplete specimens [9]. There have also been species described from Poland and, most pertinently for this study, Morocco. *Titanichthys termieri*, one of the largest members of the genus, is known from the Tafilalet basin in South Morocco [30].

The *Titanichthys* and *Tafilalichthys* specimens used in this study were found in Morocco, where the Famennian strata are known for their high quantity of preserved placoderms [31,32]. Both specimens were discovered in the Southern Maïder basin, which neighbours the Tafilalet basin. The type specimens of both *Titanichthys termieri* and *Tafilalichthys lavocati* were described in the Tafilalet basin [9], therefore the fossils in this study can be assigned to those species with some confidence.

The primary subject of this investigation was a nearly complete *Titanichthys termieri* left inferognathal (PIMUZ A/I 4716 – Figure 1). It is missing only the anterior tip, representing a small portion of the overall length – with a total length of 96 cm without the tip. While arthrodire inferognathals are typically divided antero-posteriorly into distinct dental and blade portions [33], in *Titanichthys termieri* there is a much more gradual transition between the narrow posterior blade and the thicker anterior section. The posterior blade is narrow mediolaterally and high dorsoventrally, similarly to other arthrodires [34]. The anterior ‘dental’ section seems an inapt term for a region devoid of any dentition; with no denticles or shearing surfaces visible along the jaw – a pattern common across all *Titanichthys* species with known gnathal elements [9].

Titanichthys is considered to have been a member of the family Mylostomatidae, with

Bungartius perissus and *Tafilalichthys lavocati* [8] - both of which are thought to have been durophagous, although there was little evidence of *Tafilalichthys* gnathal elements prior to this paper [26]. Durophagy seems an extremely plausible feeding method for *Bungartius*, with a thickened occlusal surface at the anterior symphyseal region on its inferognathal appearing ideally suited to function as a shearing surface [35].

To date, the only described *Tafilalichthys* jaw specimen is an anterior supragathal [30], which indicated that *Tafilalichthys* was durophagous, although not specialised to the same degree as the related *Bungartius* or *Mylostoma* [26]. The *Tafilalichthys* inferognathal investigated herein (PIMUZ A/I 4717 - Figure 2) suggests that *Tafilalichthys* may have been more adapted for durophagy than previously thought, with the anterior symphyseal region somewhat resembling that previously described for *Bungartius* and other durophagous arthrodires – with the occlusal dorsal surface partially composed of a cancellous texture [36]. However, this surface is flattened to the point of horizontality in *Tafilalichthys*, whereas both *Bungartius* [35] and *Mylostoma* [37] have more curved dental regions – which potentially could also have ‘chopped’ prey [38].

Like *Mylostoma*, the posterior ‘blade’ portion of *Tafilalichthys*’ inferognathal comprises over half of the total length, as opposed to a smaller proportion in the earlier, Frasnian (383-372 Ma) mylostomatids – which were less specialised for durophagy [39]. This proportional lengthening of the blade is thought to have increased the area of attachment for the adductor (jaw-closing) muscles, thereby increasing the bite force; crucial when specialising upon tough to digest, hard-shelled prey [40].

Dunkleosteus was selected as a comparison due to its well-documented status as an apex predator [41] and an arthrodire - indicating fairly close relatedness with Mylostomatidae [8]. Ideally, a *Dunkleosteus marsaisi* specimen could have been located, as it co-occurred with *Titanichthys termieri* in the Southern Maïder basin [42], however this did not prove possible. Instead, *Dunkleosteus terrelli*, known from the Cleveland Shale, was used. While *D. terrelli* was substantially larger than *D. marsaisi*, the skulls of the two species seem to have broadly similar shapes [9]. Given that all jaws in this study were scaled to the same length, using either species would be likely to yield similar results.

The inferognathal of *Dunkleosteus terrelli* is more clearly differentiated into blade and dental portions than the other arthrodires in this study. The dental portion is divided into an anterior fang-shaped cusp, presumably for puncturing flesh, and a posterior sharp blade which occluded with a parallel bladed surface on the supragathal [27]. This masticating, bladed surface is part of the dental portion of the inferognathal, separate from the edentulous posterior portion [34]. From a simple visual comparison, it appears much better-adapted for consuming large prey than *Titanichthys*.

Extant taxa

Sharks were selected as an extant comparison group due to the range of feeding strategies they display, including taxa with potentially analogous lifestyles to the three arthrodiran species investigated. The basking shark (*Cetorhinus maximus*) is a megaplanktivore, approaching a body length of 12 m [43]. Being closely related to an apex predator it co-occurs with, the great white shark (*Carcharodon carcharias*), the basking shark seems analogous with the proposed ecological niche of *Titanichthys*. The whale shark (*Rhincodon typus*) would potentially have represented an even closer analogue for *Titanichthys*, having also evolved from durophagous ancestors [44], as seems likely for *Titanichthys* – unfortunately whale shark specimens could not be accessed for this study.

The great white shark is an ideal analogue for *Dunkleosteus*, being a lamniform shark (the same order as *Cetorhinus* [45]) with a powerful bite force befitting of an apex predator [46]. The horn shark *Heterodontus francisci* was selected for its durophagous lifestyle [47], making it analogous for the proposed feeding strategy of *Tafilalichthys*. However, it is not that closely related to the other sharks in this study; being in a different order, the Heterodontiformes [48]. Due to the absence of known durophagous species among lamniform sharks, *Heterodontus* is the most suitable candidate for a durophage related to *Cetorhinus*.

To provide a further comparison point, and potentially assess whether certain lower jaw structural changes were common among parallel evolutionary pathways, whales were also included in the analysis. The planktivorous blue whale (*Balaenoptera musculus*) was compared with the killer whale (*Orcinus orca*), an apex predator [49]. A third comparison species was not used because of the lack of durophagous whale species. Due to the considerable evolutionary distance between the filter-feeding mysticetes and macrophagous odontocetes – which diverged around 38 Ma [50] – this comparison may be somewhat less strong. When the investigated species are co-occurring sister taxa, like *Titanichthys* and *Tafilalichthys*, morphological differences are more likely to be driven by a single explanatory factor, like divergence of function. There is a far greater possibility that differences between distantly-related species are due to a myriad of different factors, the effects of which are hard to distinguish between. Results for whales should be viewed with that caveat in mind.

Finite element model construction

All jaw models were produced using surface scans of the original specimens. Some specimens had already been scanned prior to this research (Table 1), those remaining were scanned at the University of Zurich using an Artec Eva light 3D scanner (Artec 3D). Surface scans were used instead of computerised tomography (CT) scans as the size and composition of some specimens rendered CT scanning extremely difficult. This, unfortunately, prevented the incorporation of internal features into the models; therefore, the jaws were treated as homogenous structures. Doing so has previously yielded differing results to more accurate, heterogenous models [46]. However, the surface scans should still prove valid for the purely shape-based comparison undertaken in this paper; although CT-scanning would be essential for an assessment of the absolute performance of *Titanichthys*' jaw. While the shark jaws were originally CT-scanned [51], only the surfaces were used to ensure methodological equivalence between species.

Jaw scans were processed, cleaned (removal of extraneous material and smoothing of fractures) and fused (where jaws were scanned in separate pieces) using a combination of Artec Studio 12 (Artec 3D), Avizo 9.4 (FEI Visualization Sciences Group) and MeshLab [52]. Jaw models were scaled to the same total length, as model size and forces applied had to be kept constant to ensure the analysis was solely investigating the effect of jaw shape on stress/strain resistance. Ideally, the models would have been scaled to the same surface area instead of length, as this typically produces stress comparisons of greater validity [53]. Similarly, scaling models to volume is most effective for comparing strain resistance. However, the extremely varied dentition among the various species skewed the results when models were scaled to either the same surface area or volume; an effect that has been noted previously [34]. Consequently, it was judged that equivocating model size using jaw length produced reasonably comparative models.

The muscle force applied to the jaws was adapted from a prior investigation of arthrodiran jaw mechanics [34], which primarily centred on a *Dunkleosteus terrelli* inferognathal. Consequently, all jaws were scaled to the length of the *D. terrelli* inferognathal scanned herein. The material properties proposed by Snively *et al.* [34], based on typical arthrodiran inferognathals, were applied to all jaw models. Treating each jaw as one homogenous material, jaws were assigned a Young's modulus of 20 GPa and a Poisson's ratio of 0.3. A vertical force of 300N was applied at the presumed central point of adductor mandibulae attachment. While this does

not accurately represent the force exerted by the muscles, it is a decent approximation given the absence of further skull material with which muscle action could be modelled [27]. Each jaw was constrained at the attachment point with the skull – typically on the dorsal surface at the posterior end of the lower jawbone. This constraint involved fixing a node at the attachment point for both translation and rotation in the X, Y and Z axes. Another constraint was applied to a node at the base of the anteriormost tooth (or the roughly analogous location proportionally for species with no discernible dentition), fixed for translation in the Y axis – effectively simulating the dentition being suspended within an item of prey.

Each jaw scan was ‘meshed’ – divided into elements, comprising the 3D volume of the jaw – in Hypermesh (Altair Hyperworks; Troy, Michigan), whereupon forces, constraints and material properties were applied. Each loaded model was imported into Abaqus (Dassault Systemes Simulia Corp., Providence), where FEA was performed. Every element is comprised from multiple nodes, which make up the outline of the element. Given the material properties of the model and the applied constraints, the deformation at each node can be simulated using FEA [29]. From these deformations, the stresses and strains experienced by each element within the model can be calculated.

The primary indicator selected was von Mises stress, which relates to the likelihood of ductile yielding causing a structure to fail [53]. Maximum principal stress distribution across the jaw was also analysed, as an indicator of the probability of brittle fracture. Given that bone responds in both ductile and brittle manners to stresses [54], recording both von Mises and maximum principal stress values should provide a more comprehensive profile of the jaws’ robustness. The maximum principal strain value of each element was also recorded. The extent of strain experienced within a structure indicates the degree of deformation undergone by the structure, therefore models with lower strain values are more resistant to deformation [53].

Experimentally, it was observed that proportional comparisons based on each of the three metrics produced extremely similar results (see Supplementary). Consequently, only von Mises stress was used for further analysis, as it seemed to reflect structural robustness effectively.

It is important to emphasise that the values displayed herein are very unlikely to accurately represent the actual values of stress the jaws would experience. Re-scaling of the jaw length, as well as assignment of equal material properties and applied forces, renders the absolute values irrelevant. Instead, these measures all served to validate comparisons between the different finite element models. Consequently, it is the proportional differences between the stress values experienced across the respective jaws that should be the main focus of analysis, as the disparities observed will indicate the relative robustness of the jaw shapes.

Finite element analysis

Initial comparison of stress distribution across the finite element models will be purely visual, which has been used repeatedly to effectively distinguish mandibles by their dietary function [34,55]. This will enable qualitative assessment of the stress patterns in the respective jaws, highlighting regions of particularly high stress and enabling an approximation of the differing overall resistances to stress.

In order to compare between the jaws quantitatively, average von Mises stress values were recorded for each model, from every element across the model. Typically, mean values are used [56], but median values may prove more robust to being skewed by extreme values [57]. Consequently, both mean and median values were calculated for *Titanichthys*, whereupon the value of the respective metrics could be assessed. Averaging has the advantage of enabling comparison of total stress and strain resistance with a far greater degree of precision than from pure visual comparison [58]. When combined with visual comparison, particularly weak or robust sections of the structure can still be identified. In addition, Kruskal-Wallis tests were used to assess for significance in any disparities between species’ median von Mises stress values [59] – although the massive sample sizes of the underlying data, with some models having over 200,000 elements, are likely to imbue even small differences with statistical significance.

However, averaging results can be skewed by element size, with smaller elements typically yielding more accurate results [60]. To combat this, an ‘Intervals method’ has been proposed [58], which incorporates element volumes to provide a more valid result. This method could allow for considerably more effective comparison of finite element models, and consequently more precise distinction between feeding strategies.

Intervals Method

The full method is described in the original paper [58], but will be outlined in brief here. Following FEA, all elements in the model are sorted by their von Mises stress value. These are then grouped into a number of

'intervals', each of which has an equal range of stress values. 50 intervals proved the optimal amount in the original experiment, so are used in this test (however as few as 15 intervals were still broadly effective at discriminating between dietary functions).

The cumulative volume of the elements represented in each of the 50 intervals can be calculated, then represented as a percentage of the total model volume. This represents the distribution of stresses across a model, characterising the proportion of the elements experiencing particular stress levels. A principal component analysis (PCA) is performed, based on the percentage of jaw volume represented in each interval, plotting the jaw models on two axes (principal components) that should describe the majority of variation in stress distribution. Species with similar diets should group together to an extent, if the differences in stress distribution between feeding strategies can be categorised. This method has successfully distinguished between different dietary preferences in jaw models previously, although using species within the same genus [58], much more closely related than the species tested here. Experimentally, it was observed that the whale jaw models were poorly-suited for direct comparison with the other species, as the considerable morphological disparity resulted in some models being represented in less than half of the stress intervals. Consequently, whales were removed from the PCA, to prevent skewing of the results.

4. Results

Visual comparison

The magnitudes of von Mises stress vary significantly between the placoderm inferognathals, but the general stress distribution patterns are relatively consistent (Figure 3). The highest von Mises stress values for all three species occur in the posterior bladed region; particularly close to the jaw attachment point, likely as a result of the constraint applied there. Higher stress values are experienced on the lateral aspects of each jaw than the medial. The fixed anterior point is also associated with high stress, but these regions are much more localised than at the jaw and muscle attachment points. *Titanichthys* exhibits the least resistance to von Mises stress among the placoderms, with *Dunkleosteus* proving the most resistant.

Visually, *Carcharodon* appears to be the least resistant to von Mises stress of the three shark lower jaws (Figure 3), with *Heterodontus* likely the most resistant. In whales, the mandible of *Orcinus* is clearly more resistant than *Balaenoptera*, which is characterised by extremely high levels of von Mises stress, experienced across the majority of the structure (Figure 3).

Quantitative comparison

Averaging the per element von Mises stress values produced differing results depending on whether the median or mean was used. However, while the actual values produced diverged (Table 2), the proportional differences between the species remained relatively consistent. Consequently, either method seems equally applicable; to simplify the results, the median will be used as the method of averaging henceforth.

Among the placoderms, the inferognathal of *Titanichthys* was the least resistant by some margin (Figure 4). The median elemental von Mises stress value for the inferognathal of *Tafilalichthys* represented 71% of the equivalent figure for *Titanichthys*, while in *Dunkleosteus* it was just 37%.

In general, the average von Mises stress values for shark jaws (Figure 4) were lower than in placoderms, with the highest value in sharks (in *Cetorhinus*) only slightly (0.1 MPa) higher than the lowest value in placoderms – for *Dunkleosteus*. While the jaws are typically more robust in sharks, there are some similar patterns when comparing proportional differences between the sharks. The filter-feeding basking shark displays the highest average stress, although the difference between it and the macropredatory great white shark is notably smaller than the (potentially) corresponding disparity between *Titanichthys* and *Dunkleosteus*. The median von Mises stress for *Carcharodon* is 75% of the equivalent for *Cetorhinus*, a disparity dwarfed by the much greater resistance to stress observed in *Heterodontus* (28% of *Cetorhinus*).

With a median von Mises stress value of 9.32 MPa, the mandible of *Balaenoptera musculus* is markedly less resistant to stress than all other jaws investigated (Figure 4). There is a large inter-lineage disparity with the von Mises stress resistance of *Orcinus*, the median of which is 19% of that of *Balaenoptera*.

Kruskal-Wallis tests revealed the differences between the median von Mises stress values for each species to be highly significant ($p < 0.0001$).

Intervals method

The intervals method PCA (Figure 5) attempts to differentiate between species based on the distribution of stress across the jaw models. The method groups *Titanichthys* with the planktivorous *Cetorhinus* and the closely related *Tafilalichthys*. There is little obvious diet-based grouping of the macrophagous species, with the non-*Cetorhinus* shark species relatively close together.

5. Discussion

Titanichthys' jaw conforms to a suspension feeding ecology

The inferognathal of *Titanichthys* was less resistant to von Mises stress than those of either *Dunkleosteus* or *Tafilalichthys*. *Dunkleosteus terrelli* has been substantively established as an apex predator [27,41], while *Tafilalichthys* can be considered to have been durophagous with some confidence, due to its morphological resemblance to, and close relatedness with, known durophagous arthrodires [8,26]. The comparatively high levels of stress observed in the inferognathal of *Titanichthys* suggest that neither feeding strategy would have been possible for *Titanichthys*, as its inferognathal would likely have failed (either by ductile yielding or brittle fracture) if exposed to the forces associated with the alternate feeding strategies. This strongly suggests that it was indeed a filter-feeder, as predicted based on its jaw morphology.

If *Titanichthys* were a filter-feeder, the primary function of its jaw would have been to maximise the water taken into the oral cavity during feeding, thereby increasing the rate of prey intake [61]. Morphologically, the inferognathal of *Titanichthys* seems ideally suited for this purpose – its elongation increased the maximum capacity of the oral cavity, which correlates with water filtration rate [62]. The perceived elongation of *Titanichthys*' inferognathal can be demonstrated by comparing its size with an inferognathal of the similarly-sized *Dunkleosteus terrelli*, which is clearly wider and shorter - the specimen used here is less than half the length of the corresponding *Titanichthys* inferognathal [63]. The narrowing of *Titanichthys*' inferognathal, associated with elongation, would have reduced its mechanical robustness (as displayed in this study). This adaptation would likely be unfeasible for a species reliant upon consuming large or hard-shelled prey, as it would result in a fitness reduction from an adaptive peak [64]. Should planktivory have been possible for *Titanichthys*, it could have been freed from evolutionary constraints preventing any weakening of its inferognathal; replaced by selection for maximising prey intake.

While the inferognathals of both species were considerably more mechanically resilient than that of *Titanichthys*, there is still a sizable disparity between the von Mises stress values observed in *Dunkleosteus terrelli* and *Tafilalichthys lavocati*. Biomechanical analyses have suggested that *Dunkleosteus* was capable of feeding on both highly-mobile and armoured prey [27], due to its high bite force and rapid jaw kinematics. In rodents, species with generalist diets have been shown to be more resistant to stresses across the skull than their more specialist relatives [65]. It is possible the comparatively generalist *Dunkleosteus* had a more stress-resistant inferognathal than the specialist durophage *Tafilalichthys* for the same reason.

In sharks, the highest values of stress are seen in the filter-feeding basking shark. This adds weight to the conclusion that *Titanichthys* was a filter-feeder, as the obligate planktivorous shark is significantly less resistant to stress than its durophagous and macropredatory relatives. The disparity in stress resistance between the lower jaws of *Carcharodon* and *Cetorhinus* is smaller than the equivalent disparity between *Titanichthys* and *Dunkleosteus*. The basking shark's lower jaw retains the same basic structure, albeit with less complexity, of the other shark species; whereas the lower jaw of *Titanichthys* is more morphologically divergent from the other placoderm species investigated, probably causing the more disparate results. It is notable that, while there is a large difference in lower jaw robustness between *Cetorhinus* and *Titanichthys* (median von Mises stress of 0.78 MPa compared with 1.83 MPa, respectively), *Carcharodon* and *Dunkleosteus* performed very similarly. The median von Mises stress for *Carcharodon* was 85% of the respective value for *Dunkleosteus*. *Dunkleosteus* likely occupied the equivalent niche as *Carcharodon*, but their methods of subduing prey probably differed as a result of very efficient locomotion in the great white shark [66], which is unlikely to have been replicated in the heavy, less streamlined *Dunkleosteus* [67]. Consequently, the great white shark lower jaw was expected to prove more resistant to stress than the inferognathal of *Dunkleosteus* – and this may have been seen to a greater extent if cartilaginous properties were applied to the shark jaw as homogenous bone has previously resulted in underestimated stress resistance [46] and a lower Young's modulus associated with calcified cartilage would result in higher jaw strain. On the other hand,

prior research indicating that the bite force: body mass ratio of *Dunkleosteus* is roughly equivalent to that of the great white shark [27] suggests that similar stress resistances, when scaled to length, are to be expected. The stress resistance of the great white shark's lower jaw may have been roughly equivalent to that of *Dunkleosteus*, but no such resemblance between potential analogues was observed in the durophagous species. The lower jaw of *Heterodontus francisci* is a thick structure devoid of ornamentation beyond its dentition, which proved to be substantially more robust than the lower jaw of any other species investigated. The mass-specific bite force of *Heterodontus francisci* has been shown to markedly exceed that of *Carcharodon carcharias* [68], enabling efficient crushing of its hard-shelled prey. Consequently, the disparity in stress resistance between the two species is not unexpected.

What initially seems more surprising is the even larger difference between the jaw robustness of the two durophagous species, with the horn shark being far more resilient than *Tafilalichthys*. Their roughly equivalent diets would suggest similar mechanical requirements of their jaws; however, the disparity may be explained by behavioural differences. Durophagous placoderms are thought to have primarily broken down the hard shells of their prey utilising shearing, as opposed to the more mechanically taxing, crushing mechanism seen in chondrichthyans and other post-Devonian fish [69]. This suggests that the jaws of *Tafilalichthys* would have experienced less stress than those of the shell-crushing *Heterodontus* [47].

It is worth noting that the shark finite element models were produced using surface scans originally created for use in a geometric morphometric study [51]. Consequently, they were not ideally suited to being discretised into a single, uniform surface. Despite extensive remeshing using both Blender [70] and Hypermesh, the shark jaw models were still of poorer quality than the other jaw models. The impact of this on the overall results are difficult to determine, but it should be kept in mind that the broad patterns are of more use than any specific numerical values.

The fundamental pattern outlined within this study is demonstrated further in whales: the mandible of the filter-feeding blue whale is less resistant to von Mises stress than that of the macropredatory killer whale, but with a far greater disparity than in the other lineages. Jaw elongation is seen to a far greater degree in the mysticete whales than the other megaplanktivorous lineages investigated, probably as a consequence of the energetically-expensive 'lunge feeding' method utilised by most mysticetes [71]. This is doubly true for the massive jaws of the blue whale, which enable incredibly efficient feeding despite substantial mechanical expenditure [72]. Consequently, resistance to stress may be lowest in the blue whale jaw as a result of maximising feeding efficiency.

The mandible of *Orcinus* is considerably more resistant to stress than that of *Balaenoptera*, but the median values are still notably higher than in *Carcharodon* and *Dunkleosteus*, the proposed analogues of the killer whale. Ecological reasons for this are difficult to determine, with the typical diet of an orca resembling the diet of a great white shark: centering on marine mammals [73] but sufficiently generalist to predate a wide range of species [74]. This ecological similarity would seem to suggest roughly equivalent jaw robustness, a pattern, which is not seen here.

Methodological factors may have impacted the modelling results for the whale jaws. The orca's teeth were not attached to the scanned mandible. Teeth were generally associated with relatively low stress values in this study, removing these regions from the model may have raised the average values. Manually attaching them to the digital model was considered, but the imperfect nature of this would probably have further reduced the validity of the model; similarly, removing dentition of basking shark jaws or the bone parts used for cutting or crushing in placoderm jaws would have been impossible.

All jaw scans were scaled to the same length, to circumvent the impact of teeth on scaling to the same surface area. While this seemed to improve the validity of comparisons between the model placoderm and shark jaws, it may have had the opposite effect with the whale jaws. Scaling the blue whale jaw rendered it extremely narrow relative to the other jaws, to an unrealistic extent. This may partially explain the average stress value calculated in the blue whale jaw massively exceeding those of any other species. Indeed, when the whale jaws were scaled to the same surface area, the average stress values in the orca's jaw were around 70% of the equivalent values in the blue whale. By contrast, re-scaling had little impact on the orca's jaw robustness compared with the other apex predators. Again, the large-scale trends are much more valuable than any specific numerical values – and using either method revealed that the megaplanktivore jaw was significantly less mechanically resilient than that of the apex predator.

The intervals method is probably better-suited to comparing between more closely-related species [58], as their morphology would likely be more homogenous – making function-related divergences more central to

the analysis. Despite the vast evolutionary distances involved, the method effectively grouped the planktivorous *Cetorhinus* together with *Titanichthys*. This may suggest that *Titanichthys* was a filter-feeder, as the jaws of *Titanichthys* and *Cetorhinus* are very distinct morphologically, yet consistent patterns in von Mises stress distribution between them are statistically quantifiable. The close placement of *Titanichthys* and *Tafilalichthys* does suggest some caution should be taken with any interpretation, although this is likely more a function of their close relatedness than of a shared ecological niche. The PCA did not group the durophagous or macropredatory species together, although this was predictable to an extent as the stress values of those species seemed to be more influenced by their lineage than their diet. Despite this, the intervals method's detection of corresponding stress patterns between (potentially) filter-feeding species is notable. This method should be applied in a variety of contexts moving forward, to assess for mechanical adaptations underlying functional divergences in other lineages.

Trajectories in megaplanktivory – durophagous origins?

Complete *Tafilalichthys* inferognathals are not previously figured in the literature. Consequently, the specimen described in this study is significant for advancing our understanding of arthrodiran interrelationships and the evolutionary pathway that seemingly resulted in obligate planktivory in *Titanichthys*. The morphology and mechanical performance of the inferognathal both indicate that durophagy was the most likely feeding strategy for *Tafilalichthys*, supporting its proposed phylogenetic position within the Mylostomatidae [8]. With all the major mylostomatids (excluding *Titanichthys*) likely to have been durophagous, it seems reasonable to conclude that *Titanichthys* evolved from durophagous ancestors.

Evolutionary transitions from durophagy to planktivory have occurred a number of times. The filter-feeding whale shark (*Rhincodon typus*) arose from the typically benthic Orectolobiformes [44]. Its closest relative, the nurse shark (*Ginglymostoma cirratum*), is durophagous, feeding principally on hard-shelled invertebrates [75]. Similarly the sister taxon of the planktivorous Mobulidae (manta and devil rays) are the durophagous Rhinopteridae (cownose rays) [76,77]. In a less clear parallel, the only pinniped proposed to have been durophagous [78] was relatively closely related to the ancestor of the Lobodontini- pinnipeds uniquely specialised for planktivory [79].

Temporal evolution and extinction of megasuspension feeders

The emergence of a megaplanktivore in the Famennian may hold similar clues to the degree and nature of marine primary productivity during the Devonian period [80]. Modern forms migrate to regions of high seasonal productivity, such as mysticetes seeking arctic oceans and highly productive upwelling zones. Basking sharks focus on relatively less productive seasonal blooms in shallow boreal and warm temperate waters, while whale sharks are associated to tropical waters and seasonal blooms and spawning events in this realm. The evolution of megaplanktivores coincided with periods of high productivity [15,16]. For example, the radiation of mysticete whales coincide with the Neogene cooling pump and onset of the circumantarctic polar current, resulting in a stronger thermohaline pump. It has been noted that the emergence of suspension feeding pachycormid fishes correlate with the evolution of key phytoplankton: Dinoflagellates, diatoms and coccolithosphorids could reflect the increase in primary productivity that led to the Mesozoic Marine revolution [81]. While perhaps not necessarily being drivers of the revolution, the conditions permissive of such a radiation in marine primary producers may indeed reflect a marked shift in opportunity. Similarly, suspension feeding radiodonts during the Cambrian explosion [4,82,83] radiated synchronously with the first establishment of a tiered food chain with several (at least four) levels of consumers. While the Cambrian radiation may be entirely unique with the innovation of micropredation [84], evidence for increased primary productivity is manifested in global early Cambrian phosphate deposits [85] often associated to upwelling systems in modern oceans [86].

The Devonian saw the first emergence of arborescent plants on land [23]. This resulted in deeper rooting systems, higher silicate rock weathering and nutrient run off into the oceans. While increasing primary productivity, it also led to near global deep ocean anoxia, black shale deposition [87,88] and the Frasnian-Famennian Kellwasser event, one of the 'big five' mass extinctions [89]. The increased nutrients going into circulation may well have been the necessary push for allowing arthrodires to explore this ecological niche of megaplanktivory as the first vertebrates on record.

The apparent punctuation and compelling correlation between major marine radiations, shifts in apparent productivity and megaplanktivores may be of interest for understanding how this unique ecological strategy

respond to global perturbations, such as human induced climate change. With their potential added sensitivity, megaplanktivores may be 'canary birds' for ocean ecosystem health. Some caution is advised, however. There may be taphonomic biases preventing the recognition of each and every megaplanktivore in existence at a given time. As with *Titanichthys*, tell-tale features of ecology may have been lost during fossilisation. As a rule one would want to have the filter feeding apparatus preserved, but otherwise other associated anatomical adaptations or stomach contents will need to be identified [90]. There are almost certainly other planktivorous species in the fossil record yet to be identified, shown by the recent re-appraisal of the Cretaceous plesiosaur *Morturneria seymourensis* as a probable filter-feeder [91]. Indeed, there are even other placoderms that may have been planktivorous: the arthrodire *Homostius* had a narrow jaw devoid of dentition or shearing surfaces and substantially pre-dated *Titanichthys* [92]. The common reduction in stress/strain resistance observed here could be used as an indicator of planktivory in such cases where it seems plausible but cannot be identified definitively, due to the absence of fossilised filtering structures.

6. Conclusion

Finite element analysis of the lower jaw of *Titanichthys* revealed that it was significantly less resistant to von Mises stress than those of related arthrodiras that utilised macrophagous feeding strategies. This suggests that these strategies would not have been viable for *Titanichthys*, as its jaw would have been insufficiently mechanically robust. Consequently, it is highly likely that *Titanichthys* was a filter-feeder – a feeding method that is likely to exert considerably less stress on the jaw than macrophagous feeding modes. The validity of assigning filter-feeding based on jaw mechanical resilience is supported by the roughly equivalent patterns known from lineages containing extant filter-feeders.

Common morphological trends in the convergent evolution of megaplanktivores can be not only observed but quantified mechanically utilising finite element analysis. A variety of methods were used to compare between the jaw models, due to imperfections with solely comparing visually or using average stress. The intervals method grouped feeding strategies to an extent, providing an additional perspective.

Tafilalichthys, a member of the Mylostomatidae and probably one of *Titanichthys*' closest relatives, appears to have been durophagous. Durophagy is the likely feeding mode of all crown-group mylostomatids except *Titanichthys*, suggesting that it evolved from a durophagous ancestor. This durophage-to-planktivore transition is surprisingly common among convergently-evolved giant filter-feeders: also seen in multiple, independently-evolved planktivorous elasmobranch lineages.

The presence of a megaplanktivore in the Famennian supports the theory that productivity was high in the Late Devonian, likely a result of increased eutrophication caused by the diversification of terrestrial tracheophytes and the advent of arborescence. It reflects the link between the increasing complexity of Devonian marine ecosystems and functional diversity of Arthrodira, which occupied a wide range of ecological niches. Most significantly, it reveals that vertebrate megaplanktivores likely existed over 150 Ma prior to the Mesozoic pachycormids, previously considered the earliest definitive giant filter-feeders.

Acknowledgments

We would like to thank Jordi Marcé-Nogué, for allowing us to utilise his technique and adjusting it for our data. Jaw scans were provided by the Natural History Museum in London, the Idaho Museum of Natural History and Matt Friedman; others were acquired from Pepijn Kamminga's online repository of shark jaw models [49] – we extend our gratitude to them all. We also thank James Boyle for his insight and translation of relevant literature.

Ethical Statement

This article does not present research with ethical considerations. No live animals were used in the study.

Funding Statement

CK was supported by the Swiss National Science Foundation SNF (project nr. 200020_184894).

Data Accessibility

The datasets supporting this article have been uploaded as part of the Supplementary Material.

Competing Interests

We declare that we have no competing interests.

Authors' Contributions

S.J.C, E.J.R and J.V designed the study. C.K provided and scanned the Moroccan placoderm specimens. S.J.C carried out all analysis and wrote the manuscript, with critical revision from all authors. All the authors gave final approval for submission.

References

- Sanderson SL, Wassersug R. 1990 Suspension-Feeding Vertebrates. *Sci. Am.* **262**, 96–102. (doi:10.2307/24996794)
- Costa DP. 2009 Energetics. *Encycl. Mar. Mamm.*, 383–391. (doi:10.1016/B978-0-12-373553-9.00091-2)
- Clapham P. 2001 Why do Baleen Whales Migrate?. *Mar. Mammal Sci.* **17**, 432–436. (doi:10.1111/j.1748-7692.2001.tb01289.x)
- Vinther J, Stein M, Longrich NR, Harper DAT. 2014 A suspension-feeding anomalocarid from the Early Cambrian. *Nature* **507**, 496–499. (doi:10.1038/nature13010)
- Friedman M, Shimada K, Martin LD, Everhart MJ, Liston J, Maltese A, Triebold M. 2010 100-million-year dynasty of giant planktivorous bony fishes in the Mesozoic seas. *Science* **327**, 990–3. (doi:10.1126/science.1184743)
- Carr RK. 1995 Placoderm diversity and evolution. *Bull. du Muséum Natl. d'Histoire Nat. 4ème série – Sect. C – Sci. la Terre, Paléontologie, Géologie, Minéralogie* **17**, 85–125.
- Percival LME, Davies JHFL, Schaltegger U, De Vleeschouwer D, Da Silva A-C, Föllmi KB. 2018 Precisely dating the Frasnian–Famennian boundary: implications for the cause of the Late Devonian mass extinction. *Sci. Rep.* **8**, 9578. (doi:10.1038/s41598-018-27847-7)
- Boyle J, Ryan MJ. 2017 New information on *Titanichthys* (Placodermi, Arthrodira) from the Cleveland Shale (Upper Devonian) of Ohio, USA. *J. Paleontol.* **91**, 318–336. (doi:10.1017/jpa.2016.136)
- Denison RH. 1978 *Placodermi*. Volume 2. München.
- Sanderson SL, Wassersug R. 1993 Convergent and alternative designs for vertebrate suspension feeding. In *The Skull* (eds J Hanken, BK Hall), pp. 37–112. University of Chicago Press.
- Schumacher BA, Shimada K, Liston J, Maltese A. 2016 Highly specialized suspension-feeding bony fish *Rhinconichthys* (Actinopterygii: Pachycormiformes) from the mid-Cretaceous of the United States, England, and Japan. *Cretac. Res.* **61**, 71–85. (doi:10.1016/J.CRETRES.2015.12.017)
- Liston J. 2013 The plasticity of gill raker characteristics in suspension feeders: Implications for Pachycormiformes. In *Mesozoic Fishes 5 - Global Diversity and Evolution* (eds G Arratia, HP Schultze, MVH Wilson), pp. 121–143. München: Verlag Dr. Friedrich Pfeil.
- Brazeau MD, Friedman M, Jerve A, Atwood RC. 2017 A three-dimensional placoderm (stem-group gnathostome) pharyngeal skeleton and its implications for primitive gnathostome pharyngeal architecture. *J. Morphol.* **278**, 1220–1228. (doi:10.1002/jmor.20706)
- Berger WH. 2007 Cenozoic cooling, Antarctic nutrient pump, and the evolution of whales. *Deep Sea Res. Part II Top. Stud. Oceanogr.* **54**, 2399–2421. (doi:10.1016/J.DSR2.2007.07.024)
- Marx FG, Uhen MD. 2010 Climate, critters, and cetaceans: Cenozoic drivers of the evolution of modern whales. *Science* **327**, 993–6. (doi:10.1126/science.1185581)
- Pimiento C, Cantalapiedra JL, Shimada K, Field DJ, Smaers JB. 2019 Evolutionary pathways toward gigantism in sharks and rays. *Evolution (N. Y.)* **73**, 588–599. (doi:10.1111/evo.13680)
- Álvaro JJ, Ahlberg P, Axheimer N. 2010 Skeletal carbonate productivity and phosphogenesis at the lower–middle Cambrian transition of Scania, southern Sweden. *Geol. Mag.* **147**, 59. (doi:10.1017/S0016756809990021)
- Martin RE. 1996 Secular Increase in Nutrient Levels through the Phanerozoic: Implications for Productivity, Biomass, and Diversity of the Marine Biosphere. *Palaios* **11**, 209. (doi:10.2307/3515230)
- Niklas KJ, Tiffney BH, Knoll AH. 1983 Patterns in vascular land plant diversification. *Nature* **303**, 614–616. (doi:10.1038/303614a0)
- Morris JL *et al.* 2015 Investigating Devonian trees as geo-engineers of past climates: linking palaeosols to palaeobotany and experimental geobiology. *Palaeontology* **58**, 787–801. (doi:10.1111/pala.12185)
- Berner RA. 1997 The Rise of Plants and Their Effect on Weathering and Atmospheric CO₂. *Science (80-)* **276**, 544–546. (doi:10.1126/science.271.5252.1105)
- Le Hir G, Donnadiou Y, Goddérés Y, Meyer-Berthaud B, Ramstein G, Blakey RC. 2011 The climate

- change caused by the land plant invasion in the Devonian. *Earth Planet. Sci. Lett.* **310**, 203–212. (doi:10.1016/j.epsl.2011.08.042)
23. Algeo TJ, Scheckler SE. 1998 Terrestrial-marine teleconnections in the Devonian: links between the evolution of land plants, weathering processes, and marine anoxic events. *Philos. Trans. R. Soc. B Biol. Sci.* **353**, 113–130. (doi:10.1098/rstb.1998.0195)
24. Bambach RK. 1999 Energetics in the global marine fauna: A connection between terrestrial diversification and change in the marine biosphere. *Geobios* **32**, 131–144. (doi:10.1016/S0016-6995(99)80025-4)
25. Fletcher TM, Janis CM, Rayfield EJ. 2010 Finite Element Analysis of Ungulate Jaws: Can mode of digestive physiology be determined? *Palaeontol. Electron.* **13**, 15.
26. Lelièvre H. 1991 New information on the structure and the systematic position of *Tafilalichthys lavocati* (placoderm, arthrodire) from the Late Devonian of Tafilalet, Morocco. In *Early vertebrates and related problems of evolutionary biology* (eds M Chang, Y Liu, G Zhang), pp. 121–130.
27. Anderson PSL, Westneat MW. 2009 A biomechanical model of feeding kinematics for *Dunkleosteus terrelli* (Arthrodira, Placodermi). *Paleobiology* **35**, 251–269. (doi:10.1666/08011.1)
28. Rayfield EJ. 2007 Finite Element Analysis and Understanding the Biomechanics and Evolution of Living and Fossil Organisms. *Annu. Rev. Earth Planet. Sci.* **35**, 541–576. (doi:10.1146/annurev.earth.35.03.1306.140104)
29. Bright JA. 2014 A review of paleontological finite element models and their validity. *J. Paleontol.* **88**, 760–769. (doi:10.1666/13-090)
30. Lehman JP. 1956 Les Arthrodirés du dévonien supérieur du Tafilalet:(Sud marocain). *Éditions du Serv. géologique du Maroc* **129**.
31. Derycke C, Olive S, Groessens E, Goujet D. 2014 Paleogeographical and paleoecological constraints on paleozoic vertebrates (chondrichthyans and placoderms) in the Ardenne Massif: Shark radiations in the Famennian on both sides of the Palaeotethys. *Palaeogeogr. Palaeoclimatol. Palaeoecol.* **414**, 61–67. (doi:10.1016/j.palaeo.2014.07.012)
32. Frey L, Pohle A, Rücklin M, Klug C. 2019 Fossil-Lagerstätten, palaeoecology and preservation of invertebrates and vertebrates from the Devonian in the eastern Anti-Atlas, Morocco. *Lethaia*, let.12354. (doi:10.1111/let.12354)
33. Anderson PSL. 2008 Shape variation between arthrodire morphotypes indicates possible feeding niches. *J. Vertebr. Paleontol.* **28**, 961–969. (doi:10.1671/0272-4634-28.4.961)
34. Snively E, Anderson PSL, Ryan MJ. 2010 Functional and ontogenetic implications of bite stress in arthrodire placoderms. *Kirtlandia* **57**, 53–60.
35. Dunkle DH. 1947 A new genus and species of arthrodire fish from the Upper Devonian Cleveland Shale. *Sci. Publ. Clevel. Museum Nat. Hist.* **8**, 103–117.
36. Young GC. 2004 A homostiid arthrodire (placoderm fish) from the Early Devonian of the Burrinjuck area, New South Wales. *Alcheringa An Australas. J. Paleontol.* **28**, 129–146. (doi:10.1080/03115510408619278)
37. Dunkle DH, Bungart PA. 1945 A new arthrodire fish from the Upper Devonian Ohio shales. *Sci. Publ. Clevel. Museum Nat. Hist.* **8**, 85–95.
38. Anderson PSL. 2009 Biomechanics, functional patterns, and disparity in Late Devonian arthrodiras. *Paleobiology* **35**, 321–342. (doi:10.1666/0094-8373-35.3.321)
39. Hlavín WJ, Boreške JR. 1973 *Mylostoma variabile* Newberry, an Upper Devonian durophagous brachyothoracid arthrodire, with notes on related taxa. *Breviora* **412**.
40. Dunkle DH, Bungart PA. 1943 Comments on *Diplognathus mirabilis* Newberry. *Sci. Publ. Clevel. Museum Nat. Hist.* **8**, 73–84.
41. Anderson PS., Westneat MW. 2007 Feeding mechanics and bite force modelling of the skull of *Dunkleosteus terrelli*, an ancient apex predator. *Biol. Lett.* **3**, 77–80. (doi:10.1098/rsbl.2006.0569)
42. Cloutier R, Lelièvre H. 1998 Comparative study of the fossiliferous sites of the Devonian. *Prep. ministère l'Environnement la Faune, Gouv. du Québec*
43. Sims DW. 2008 Chapter 3 Sieving a Living: A Review of the Biology, Ecology and Conservation Status of the Plankton-Feeding Basking Shark *Cetorhinus Maximus*. *Adv. Mar. Biol.* **54**, 171–220. (doi:10.1016/S0065-2881(08)00003-5)
44. Goto T. 2001 Comparative Anatomy, Phylogeny and Cladistic Classification of the Order Orectolobiformes (Chondrichthyes, Elasmobranchii). *Mem. Grad. Sch. Fish. Sci. Hokkaido Univ.* **48**, 1–100.
45. SHIMADA K. 2005 Phylogeny of lamniform sharks (Chondrichthyes: Elasmobranchii) and the contribution of dental characters to lamniform systematics. *Paleontol. Res.* **9**, 55–72. (doi:10.2517/prpsj.9.55)
46. Wroe S *et al.* 2008 Three-dimensional computer analysis of white shark jaw mechanics: how hard can a great white bite? *J. Zool.* **276**, 336–342. (doi:10.1111/j.1469-7998.2008.00494.x)
47. Huber DR, Eason TG, Hueter RE, Motta PJ. 2005 Analysis of the bite force and mechanical design of the feeding mechanism of the durophagous horn shark *Heterodontus francisci*. *J. Exp. Biol.* **208**, 3553–3571. (doi:10.1242/jeb.01816)
48. Vélez-Zuazo X, Agnarsson I. 2011 Shark tales: A molecular species-level phylogeny of sharks (Selachimorpha, Chondrichthyes). *Mol. Phylogenet. Evol.* **58**, 207–217. (doi:10.1016/J.YMPREV.2010.11.018)
49. Ford JKB, Ellis GM, Olesiuk PF, Balcomb KC. 2010 Linking killer whale survival and prey abundance: food limitation in the oceans' apex predator? *Biol. Lett.* **6**, 139–142. (doi:10.1098/rsbl.2009.0468)
50. Marx FG, Fordyce RE. 2015 Baleen boom and bust: a synthesis of mysticete phylogeny, diversity and disparity. *R. Soc. Open Sci.* **2**, 140434–140434. (doi:10.1098/rsos.140434)
51. Kamminga P, De Bruin PW, Geleijns J, Brazeau MD. 2017 X-ray computed tomography library of shark anatomy and lower jaw surface models. *Sci. Data* **4**, 170047. (doi:10.1038/sdata.2017.47)
52. Cignoni P, Callieri M, Corsini M, Dellepiane M, Ganovelli F, Ranzuglia G. 2008 MeshLab: an Open-Source Mesh Processing Tool.
53. Dumont ER, Grosse IR, Slater GJ. 2009 Requirements for comparing the performance of finite element models of biological structures. *J. Theor. Biol.* **256**, 96–103. (doi:10.1016/j.jtbi.2008.08.017)
54. Shigemitsu R, Yoda N, Ogawa T, Kawata T, Gunji Y, Yamakawa Y, Ikeda K, Sasaki K. 2014 Biological-data-based finite-element stress analysis of mandibular bone with implant-supported overdenture. *Comput. Biol. Med.* **54**, 44–52. (doi:10.1016/J.COMPBIOMED.2014.08.018)
55. Serrano-Fochs S, De Esteban-Trivigno S, Marcé-Nogué J, Fortuny J, Fariña RA. 2015 Finite Element Analysis of the Cingulata

- Jaw: An Ecomorphological Approach to Armadillo's Diets. *PLoS One* **10**, e0120653. (doi:10.1371/journal.pone.0120653)
56. Lautenschlager S. 2017 Functional niche partitioning in Therizinosauria provides new insights into the evolution of theropod herbivory. *Palaeontology* **60**, 375–387. (doi:10.1111/pala.12289)
57. Dar FH, Meakin JR, Aspden RM. 2002 Statistical methods in finite element analysis. *J. Biomech.* **35**, 1155–1161. (doi:10.1016/S0021-9290(02)00085-4)
58. Marcé-Nogué J, De Esteban-Trivigno S, Püschel TA, Fortuny J. 2017 The intervals method: a new approach to analyse finite element outputs using multivariate statistics. *PeerJ* **5**, e3793. (doi:10.7717/peerj.3793)
59. Erhart P, Hylhik-Dürr A, Geisbüsch P, Kotelis D, Müller-Eschner M, Gasser TC, von Tengg-Kobligk H, Böckler D. 2015 Finite Element Analysis in Asymptomatic, Symptomatic, and Ruptured Abdominal Aortic Aneurysms: In Search of New Rupture Risk Predictors. *Eur. J. Vasc. Endovasc. Surg.* **49**, 239–245. (doi:10.1016/J.EJVS.2014.11.010)
60. Bright JA, Rayfield EJ. 2011 The Response of Cranial Biomechanical Finite Element Models to Variations in Mesh Density. *Anat. Rec. Adv. Integr. Anat. Evol. Biol.* **294**, 610–620. (doi:10.1002/ar.21358)
61. Sims DW. 1999 Threshold foraging behaviour of basking sharks on zooplankton: life on an energetic knife-edge? *Proc. R. Soc. B Biol. Sci.* **266**, 1437–1443. (doi:10.1098/rspb.1999.0798)
62. Goldbogen JA, Potvin J, Shadwick RE. 2010 Skull and buccal cavity allometry increase mass-specific engulfment capacity in fin whales. *Proceedings. Biol. Sci.* **277**, 861–8. (doi:10.1098/rspb.2009.1680)
63. Anderson PSL, Friedman M, Brazeau MD, Rayfield EJ. 2011 Initial radiation of jaws demonstrated stability despite faunal and environmental change. *Nature* **476**, 206–209. (doi:10.1038/nature10207)
64. Hansen TF. 2012 Adaptive Landscapes and Macroevolutionary Dynamics. In *The adaptive landscape in evolutionary biology* (eds EI Svensson, R Calsbeek), pp. 205–226. Oxford University Press.
65. Cox PG, Rayfield EJ, Fagan MJ, Herrel A, Pataky TC, Jeffery N. 2012 Functional Evolution of the Feeding System in Rodents. *PLoS One* **7**, e36299. (doi:10.1371/journal.pone.0036299)
66. Donley JM, Sepulveda CA, Konstantinidis P, Gemballa S, Shadwick RE. 2004 Convergent evolution in mechanical design of lamnid sharks and tunas. *Nature* **429**, 61–65. (doi:10.1038/nature02435)
67. Carr RK. 2010 Paleoeecology of *Dunkleosteus terrelli* (Placodermi: Arthrodira). *Kirtlandia, Clevel. Museum Nat. Hist.* **57**, 36–55.
68. Kolmann MA, Huber DR, Motta PJ, Grubbs RD. 2015 Feeding biomechanics of the cownose ray, *Rhinoptera bonasus*, over ontogeny. *J. Anat.* **227**, 341–351. (doi:10.1111/joa.12342)
69. Brett CE. 2003 Durophagous Predation in Paleozoic Marine Benthic Assemblages. In *Predator—Prey Interactions in the Fossil Record* (eds PH Kelley, M Kowalewski, TA Hansen), pp. 401–432. Boston, MA: Springer US. (doi:10.1007/978-1-4615-0161-9_18)
70. Zoppè M, Porozov Y, Andrei R, Cianchetta S, Zini MF, Loni T, Caudai C, Callieri M. 2008 Using Blender for molecular animation and scientific representation.
71. Goldbogen JA *et al.* 2012 Scaling of lunge-feeding performance in rorqual whales: mass-specific energy expenditure increases with body size and progressively limits diving capacity. *Funct. Ecol.* **26**, 216–226. (doi:10.1111/j.1365-2435.2011.01905.x)
72. Goldbogen JA, Calambokidis J, Oleson E, Potvin J, Pyenson ND, Schorr G, Shadwick RE. 2011 Mechanics, hydrodynamics and energetics of blue whale lunge feeding: efficiency dependence on krill density. *J. Exp. Biol.* **214**, 131–46. (doi:10.1242/jeb.048157)
73. Ford J, Ellis G. 2006 Selective foraging by fish-eating killer whales *Orcinus orca* in British Columbia. *Mar. Ecol. Prog. Ser.* **316**, 185–199. (doi:10.3354/meps316185)
74. Ford JKB. 2009 Killer Whale: *Orcinus orca*. In *Encyclopedia of Marine Mammals* (eds WF Perrin, B Würsig, JGM Thewissen), pp. 650–657. Academic Press. (doi:10.1016/B978-0-12-373553-9.00150-4)
75. Matott MP, Motta PJ, Hueter RE. 2005 Modulation in Feeding Kinematics and Motor Pattern of the Nurse Shark *Ginglymostoma cirratum*. *Environ. Biol. Fishes* **74**, 163–174. (doi:10.1007/s10641-005-7435-3)
76. Collins AB, Heupel MR, Hueter RE, Motta PJ. 2007 Hard prey specialists or opportunistic generalists? An examination of the diet of the cownose ray, *Rhinoptera bonasus*. *Mar. Freshw. Res.* **58**, 135. (doi:10.1071/MF05227)
77. Dunn KA, McEachran JD, Honeycutt RL. 2003 Molecular phylogenetics of myliobatiform fishes (Chondrichthyes: Myliobatiformes), with comments on the effects of missing data on parsimony and likelihood. *Mol. Phylogenet. Evol.* **27**, 259–270. (doi:10.1016/S1055-7903(02)00442-6)
78. Amson E, de Muizon C. 2014 A new durophagous phocid (Mammalia: Carnivora) from the late Neogene of Peru and considerations on monachine seals phylogeny. *J. Syst. Palaeontol.* **12**, 523–548. (doi:10.1080/14772019.2013.799610)
79. Koretsky IA, Rahmat S., Peters N. 2014 Remarks on Correlations and Implications of the Mandibular Structure and Diet in Some Seals (Mammalia, Phocidae). *Vestn. Zool.* **48**, 255–268. (doi:https://doi.org/10.2478/vzoo-2014-0029)
80. Klug C, Kröger B, Kiessling W, Mullins GL, Servais T, Frýda J, Korn D, Turner S. 2010 The Devonian nekton revolution. *Lethaia* **43**, 465–477. (doi:10.1111/j.1502-3931.2009.00206.x)
81. Knoll AH, Follows MJ. 2016 A bottom-up perspective on ecosystem change in Mesozoic oceans. *Proc. R. Soc. B Biol. Sci.* **283**. (doi:10.1098/rspb.2016.1755)
82. Van Roy P, Daley AC, Briggs DEG. 2015 Anomalocaridid trunk limb homology revealed by a giant filter-feeder with paired flaps. *Nature* **522**, 77–80. (doi:10.1038/nature14256)
83. Lerosey-Aubril R, Pates S. 2018 New suspension-feeding radiodont suggests evolution of microplanktivory in Cambrian macronekton. *Nat. Commun.* **9**. (doi:10.1038/s41467-018-06229-7)
84. Sperling EA, Frieder CA, Raman A V., Girguis PR, Levin LA, Knoll AH. 2013 Oxygen, ecology, and the Cambrian radiation of animals. *Proc. Natl. Acad. Sci. U. S. A.* **110**, 13446–13451. (doi:10.1073/pnas.1312778110)
85. Cook PJ, Shergold JH. 1986 Proterozoic and Cambrian phosphorites-nature and origin. In *Proterozoic and Cambrian phosphorites* (eds PJ Cook, JH Shergold), pp. 369–386. Cambridge Univ. Press.
86. Parrish JT. 1987 Palaeo-upwelling and the distribution of organic-rich rocks. *Geol. Soc. Spec. Publ.* **26**, 199–205. (doi:10.1144/GSL.SP.1987.026.01.12)
87. Lu M, Lu YH, Ikejiri T, Hogancamp N, Sun Y, Wu Q, Carroll R, Cemen I, Pashin J. 2019 Geochemical

- 1
2
3
4
5
6
7
8
9
10
11
12
13
14
15
16
17
18
19
20
21
22
23
24
25
26
27
28
29
30
31
32
33
34
35
36
37
38
39
40
41
42
43
44
45
46
47
48
49
50
51
52
53
54
55
56
57
58
59
60
- Evidence of First Forestation in the Southernmost Euramerica from Upper Devonian (Famennian) Black Shales. *Sci. Rep.* **9**. (doi:10.1038/s41598-019-43993-y)
88. Marynowski L, Zatoń M, Rakociński M, Filipiak P, Kurkiewicz S, Pearce TJ. 2012 Deciphering the upper Famennian Hangenberg Black Shale depositional environments based on multi-proxy record. *Palaeogeogr. Palaeoclimatol. Palaeoecol.* **346–347**, 66–86. (doi:10.1016/j.palaeo.2012.05.02)
89. Buggisch W. 1991 The global Frasnian-Famennian »Kellwasser Event«. *Geol. Rundschau* **80**, 49–72. (doi:10.1007/BF01828767)
90. Friedman M. 2011 Parallel evolutionary trajectories underlie the origin of giant suspension-feeding whales and bony fishes. *Proc. R. Soc. B Biol. Sci.* **279**, 944–951. (doi:10.1098/rspb.2011.1381)
91. O’Keefe FR, Otero RA, Soto-Acuña S, O’gorman JP, Godfrey SJ, Chatterjee S. 2017 Cranial anatomy of *Morturneria seymourensis* from Antarctica, and the evolution of filter feeding in plesiosaurs of the Austral Late Cretaceous. *J. Vertebr. Paleontol.* **37**, e1347570. (doi:10.1080/02724634.2017.1347570)
92. Mark-Kurik E. 1992 The inferognathal in the Middle Devonian arthrodire *Homostius. Lethaia* **25**, 173–178. (doi:10.1111/j.1502-3931.1992.tb01382.x)

Tables

Table 1

Specimen Number	Species	Order	Scanning Institute
PIMUZ A/I 4716	Titanichthys termieri	Arthrodira	University of Zurich
PIMUZ A/I 4717	Tafilalichthys lavocati	Arthrodira	University of Zurich
CM6090	Dunkleosteus terrelli	Arthrodira	Cleveland Museum of Natural History
BMNH 1978.6.22.1	Cetorhinus maximus	Lamniformes	Natural History Museum, London
ZMA.PISC.108688	Heterodontus francisci	Heterodontiformes	Zoological Museum, Amsterdam
ERB 0932	Carcharodon carcharias	Lamniformes	ZNA hospital Antwerp
BMNH 1892.3.1.1	Balaenoptera musculus	Mysticeti	Natural History Museum, London
NMML-1850	Orcinus orca	Odontoceti	Idaho Museum of Natural History

Table 2

Species	Median			Mean		
	von Mises Stress	Maximum Principal Stress	Maximum Principal Strain	von Mises Stress	Maximum Principal Stress	Maximum Principal Strain
Titanichthys termieri	1.83	0.65	4.95E-05	2.72	1.43	9.43E-05
Dunkleosteus terrelli	0.68	0.31	2.17E-05	0.94	0.51	3.29E-05
Tafilalichthys lavocati	1.29	0.48	3.81E-05	1.79	0.94	6.14E-05
Cetorhinus maximus	0.78	0.36	2.52E-05	0.88	0.51	3.18E-05
Carcharodon carcharias	0.58	0.26	2.01E-05	0.82	0.48	3.03E-05
Heterodontus maximus	0.22	0.11	7.78E-06	0.30	0.18	1.11E-05
Balaenoptera musculus	9.32	3.71	2.90E-04	11.66	6.56	4.17E-04
Orcinus orca	1.78	0.62	5.27E-05	2.14	1.22	7.65E-05

Figures

1
2
3
4
5
6
7
8
9
10
11
12
13
14
15
16
17
18
19
20
21
22
23
24
25
26
27
28
29
30
31
32
33
34
35
36
37
38
39
40
41
42
43
44
45
46
47
48
49
50
51
52
53
54
55
56
57
58
59
60

1
2
3
4
5
6
7
8
9
10
11
12
13
14
15
16
17
18
19
20
21
22
23
24
25
26
27
28
29
30
31
32
33
34
35
36
37
38
39
40
41
42
43
44
45
46
47
48
49
50
51
52
53
54
55
56
57
58
59
60

Figure and table captions

Table 1

The specimens used in the study and the institutes in which they were scanned. Additional *Titanichthys* and *Tafilalichthys* specimens were observed at the University of Zurich to provide a more thorough insight into the species.

Table 2

Average elemental stress and strain values for the lower jaws of various species of placoderms, sharks and whales, calculated using finite element analysis. Both median and mean values are displayed. The unit for all values is MPa (megapascals).

Figure 1

Left inferognathal of *Titanichthys termieri* (PIMUZ A/I 4716), from the Southern Maïder basin, Morocco. The specimen is nearly complete, excluding the anteriormost tip. The inferognathal lacks both dentition and shearing surfaces. It has been glued together where fractures occurred. Photographed at the University of Zurich. Total length = 96 cm.

Figure 2

Inferognathals of *Tafilalichthys lavocati* (PIMUZ A/I 4717), from the Southern Maïder basin, Morocco. Photographed at the University of Zurich. Total length = 33cm.

Figure 3

Von Mises stress distributions in the lower jaws of selected placoderm, shark and whale species, calculated using finite element analysis (generally following the methodology of Snively *et al.* [34]).

Figure 4

Median von Mises stress values for each jaw finite element model. Bar colour corresponds with the potential ecological niche of each species.

Figure 5

Principal component analysis (PCA) visualising the variation in von Mises stress distribution between the lower jaw finite element models, as indicated by the intervals method [58]. Symbol colour is used to distinguish between clades: placoderm symbols are white and shark symbols are black. Shapes correspond with the potential ecological niche of each species. The percentage values on the axes indicate the variance explained by each principal co-ordinate. PC1 and PC2 cumulatively account for 90.6% of the total variance.

Appendix B**ROYAL SOCIETY
OPEN SCIENCE****Was the Devonian Placoderm *Titanichthys* a Filter-Feeder?**

Journal:	Royal Society Open Science
Manuscript ID	RSOS-200272
Article Type:	Research
Date Submitted by the Author:	20-Feb-2020
Complete List of Authors:	Coatham, Sam; The University of Manchester, Earth Sciences Vinther, Jakob; University of Bristol, Biological Sciences Rayfield, Emily; University of Bristol, School of Earth Sciences Klug, Christian; Universität Zürich, Paläontologisches Institut und Museum
Subject:	Palaeontology < EARTH SCIENCES, biomechanics < BIOLOGY, evolution < BIOLOGY
Keywords:	Filter-Feeding, Titanichthys, Arthrodira, Finite Element Analysis, Placodermi
Subject Category:	Earth and Environmental Science

Author-supplied statements

Relevant information will appear here if provided.

Ethics

Does your article include research that required ethical approval or permits?:

This article does not present research with ethical considerations

Statement (if applicable):

CUST_IF_YES_ETHICS :No data available.

Data

It is a condition of publication that data, code and materials supporting your paper are made publicly available. Does your paper present new data?:

Yes

Statement (if applicable):

All data has been made accessible at the Dryad Digital Repository

(<https://doi.org/10.5061/dryad.9kd51c5d6>, review URL:

https://datadryad.org/stash/share/sA4rCcK0slzvPgaU3QAzH7wwlg8_Q2JI9mLPpsCQjgg). Jaw scans

used with the permission of museums have not been made available in their raw format, but the

finite element models produced using them are accessible.

Conflict of interest

I/We declare we have no competing interests

Statement (if applicable):

CUST_STATE_CONFLICT :No data available.

Authors' contributions

This paper has multiple authors and our individual contributions were as below

Statement (if applicable):

S.J.C, E.J.R and J.V designed the study. C.K provided and scanned the Moroccan placoderm

specimens. S.J.C carried out all analysis and wrote the manuscript, with critical revision from all

authors. All the authors gave final approval for submission.

Was the Devonian Placoderm *Titanichthys* a Filter-Feeder?

Samuel J. Coatham^{1*†}, Jakob Vinther¹, Emily J. Rayfield¹ and Christian Klug²

¹ Life Sciences Building, University of Bristol, 24 Tyndall Avenue, Bristol; e-mails:

sam.coatham@postgrad.manchester.ac.uk, jakob.vinther@bristol.ac.uk, E.rayfield@bristol.ac.uk

² Paläontologisches Institut und Museum, Universität Zürich, Karl-Schmid-Strasse 4, 8006 Zürich; e-mail: chklug@pim.uzh.ch

Keywords: Filter-feeding, *Titanichthys*, Arthrodira, Devonian, comparative biomechanics

1. Summary

Large nektonic suspension feeders have evolved multiple times. The apparent trend among apex predators for some evolving into feeding on small zooplankton is of interest for understanding the associated shifts in anatomy and behaviour while the spatial and temporal distribution gives clues to an inherent relationship with ocean primary productivity and how past and future perturbations to these may impact on the different tiers of the food chain. The evolution of large nektonic suspension feeders - 'gentle giants' - occurred 4 times among chondrichthyan fishes (e.g. whale and basking sharks and manta rays) and baleen whales (mysticetes), the Mesozoic pachycormid fishes and at least twice in radiodontan stem group arthropods (Anomalocaridids) during the Cambrian Explosion. The Late Devonian placoderm *Titanichthys* has tentatively been considered to have been a megaplanktivore, primarily due to its gigantic size and narrow, edentulous jaw while no filtering apparatus have ever been reported. Here the potential for microphagy and other feeding behaviours in *Titanichthys* is assessed via a comparative study of jaw mechanics in *Titanichthys* and other placoderms with presumably differing feeding habits (macrophagy and durophagy). Finite element models of the lower jaws of *Titanichthys termieri* in comparison to *Dunkleosteus terrelli* and *Tafilalichthys lavocati* reveal considerably less resistance to von Mises stress in this taxon. Comparisons with a selection of large-bodied extant taxa of similar ecological diversity reveals similar disparities in jaw stress resistance. Our results therefore conform to the hypothesis that *Titanichthys* was a filter-feeder with jaws ill-suited for biting and crushing but well suited for gaping ram feeding.

2. Introduction

Some of the largest organisms ever to have roamed the ocean and alive today are suspension feeders. The switch to feeding on the lowest levels of the trophic pyramid is a tremendous shift in food resource [1]. While pursuing large bodied prey results in adaptations towards stealth, complex hunting behaviours and expanded sensory repertoires, suspension feeding results in a host of anatomical, migratory and behavioural modifications. Locomotory speed and energy reserves scale with body mass - enabling a migratory lifestyle in some species [2] to capitalise on seasonal periods of high food abundance [3]. Invertebrate filter-feeders are known from the Cambrian [4], giant-bodied relative to their temporal counterparts. While the first definitive vertebrate megaplanktivores occurred in the Mesozoic, within the pachycormids [5], this ecological niche may in fact have originated in the Devonian.

The arthrodire *Titanichthys* occurred in the Famennian [6], the uppermost stage of the Devonian (372-359 Ma [7]). There are multiple morphological features indicating that *Titanichthys* may have been a megaplanktivore, primarily its massive size [8]. The elongate, narrow jaws lack any form of dentition or shearing surface [9]; seemingly ill-equipped for any form of prey consumption more demanding than simply funnelling prey-laden water into the oral cavity. *Titanichthys* is also known for its small orbita (relative to its size), indicating that visual acuity may not have been that important in its predatory behaviour. This is a known feature of predation in extant filter-feeders [10], so may be further evidence of planktivory. However, the filter-feeding

*Author for correspondence (sam.coatham@postgrad.manchester.ac.uk).

†Present address: Department of Earth and Environmental Sciences, Michael Smith Building, University of Manchester, Dover Street.

1 pachycormid *Rhinconichthys* has enlarged sclerotic rings [11], bringing into question the use of reduced
2 orbitals as a diagnostic character of planktivory.

3
4 Despite the numerous physical traits shared between *Titanichthys* and other definitive giant filter-feeders,
5 planktivory in *Titanichthys* has yet to be strongly supported, due to the absence of evidence of a filtering
6 structure. If *Titanichthys* was indeed a filter-feeder, presumably it would have fed in a roughly analogous
7 manner to modern planktivorous fish, which strain prey from water exiting the oral cavity through the gills
8 using elaborate gill rakers (this was also the filtering method of planktivorous pachycormids [12]). Placoderm
9 gill arches are rarely preserved [13], so the absence of a fossil filtering structure may be an artefact of the poor
10 fossil record, or it may indicate that *Titanichthys* was not a filter-feeder.

11
12 The viability of filter-feeding in *Titanichthys* is promoted by seemingly favourable conditions in the Devonian.
13 Increases in primary productivity appear to be associated with the recurrent evolution of megaplanktivores,
14 with potential expansions of available food resources enabling larger body sizes. This has been observed in the
15 diversification of mysticetes [14,15] and the origin of most filter-feeding elasmobranch clades [16], with
16 potential further correlations in the evolution of giant planktivorous anomalocarids in the Lower Cambrian
17 [17] and pachycormids in the Jurassic [18]. Productivity probably also increased throughout the Devonian,
18 with the combination of tracheophyte proliferation [19] and the advent of arborescence [20] likely accelerating
19 the rate of chemical weathering [21]. This could have resulted in enrichment of the oceanic nutrient supply via
20 runoff [22], potentially increasing marine productivity [18]. Although there is little direct proof of this [23], the
21 rise in diversity of predators with high energetic demands [24] indicates sufficient productivity to support
22 relatively complex ecosystems. Consequently, it seems probable that productivity did increase, potentially
23 facilitating the evolution of a giant filter-feeder in the Devonian.

24
25 To assess whether *Titanichthys* was indeed a filter-feeder, we investigated the mechanical properties of its jaw
26 in order to infer function. The engineering technique finite element analysis (FEA) has previously been used to
27 effectively differentiate between the mandibles of related species with differing diets [25]. Consequently, finite
28 element models of the inferognathals of *Titanichthys termieri*, *Tafilalichthys lavocati* and *Dunkleosteus terrelli* were
29 generated and compared. *Tafilalichthys* is thought to have been durophagous (specialised to consume hard-
30 shelled prey) [26], while *Dunkleosteus* was almost certainly an apex predator [27]; representing the two most
31 plausible feeding modes for *Titanichthys* (excluding planktivory). Both species were arthrodires related to
32 *Titanichthys*, with *Tafilalichthys* more closely related – likely within the same family [8].

33
34 By digitally discretising a structure into many elements and applying loads, constraints and material
35 properties, the stress and strain experienced within each element can be calculated in FEA [28]. When viewed
36 as components of the entire structure, its resistance to stress and strain can be clearly visualised, enabling
37 functional inference. While the magnitude of stress/strain values in extinct taxa are hard to definitively
38 ascertain, comparing between models loaded in the same manner is effective for comparative studies of
39 function [29]. Therefore, the mechanics of the arthrodire inferognathals will be compared based purely on
40 their shape. Extant taxa, the lifestyles of which are far better understood, will be used as a further reference
41 point, to validate the use of jaw robustness as a proxy for feeding strategy. The sharks *Cetorhinus maximus*
42 (basking), *Carcharodon carcharias* (great white) and *Heterodontus francisci* (horn) all occupy ecological niches
43 roughly analogous to those of the placoderms studied (planktivore, apex predator and durophage,
44 respectively). In addition, the cetaceans *Balaenoptera musculus* (blue whale) and *Orcinus orca* (killer whale) will
45 serve as a further planktivore-apex predator reference; albeit with much greater evolutionary distance
46 between the species.

47
48 Comparing the jaw mechanics of definitive filter-feeders with their macrophagous relatives will provide
49 clarity regarding the implications of any differences in stress/strain patterns of the placoderm jaws, informing
50 any conclusions regarding *Titanichthys*' feeding strategy. Should *Titanichthys* have been a filter-feeder, its jaw
51 would be expected to be less mechanically robust than those of related species with diets associated with
52 greater bite forces, which would exert more stress on the jaw. Consequently, the jaw of a filter-feeder is
53 predicted to be less resistant to stress and strain than those of the compared durophagous and
54 macropredatory species.

3. Materials and Methods

Placoderm Specimens

Titanichthys specimens are mostly known from the Cleveland Shale, with remains of five different *Titanichthys* species having been found there – albeit mostly from relatively incomplete specimens [9]. There have also been species described from Poland and, most pertinently for this study, Morocco. *Titanichthys termieri*, one of the largest members of the genus, is known from the Tafilalet basin in South Morocco [30].

The *Titanichthys* and *Tafilalichthys* specimens used in this study were found in Morocco, where the Famennian strata are known for their high quantity of preserved placoderms [31,32]. Both specimens were discovered in the Southern Maïder basin, which neighbours the Tafilalet basin. The type specimens of both *Titanichthys termieri* and *Tafilalichthys lavocati* were described in the Tafilalet basin [9], therefore the fossils in this study can be assigned to those species with some confidence.

The primary subject of this investigation was a nearly complete *Titanichthys termieri* left inferognathal (PIMUZ A/I 4716 – Figure 1). It is missing only the anterior tip, representing a small portion of the overall length – with a total length of 96 cm without the tip. While arthrodire inferognathals are typically divided antero-posteriorly into distinct dental and blade portions [33], in *Titanichthys termieri* there is a much more gradual transition between the narrow posterior blade and the thicker anterior section. The posterior blade is narrow mediolaterally and high dorsoventrally, similarly to other arthrodires [34]. The anterior ‘dentitionless’ section seems an inapt term for a region devoid of any dentition; with no denticles or shearing surfaces visible along the jaw – a pattern common across all *Titanichthys* species with known gnathal elements [30].

Titanichthys is considered to have been a member of the family Mylostomatidae, with

Bungartius perissus and *Tafilalichthys lavocati* [8] - both of which are thought to have been durophagous, although there was little evidence of *Tafilalichthys* gnathal elements prior to this paper [26]. Durophagy seems an extremely plausible feeding method for *Bungartius*, with a thickened occlusal surface at the anterior symphyseal region on its inferognathal appearing ideally suited to function as a shearing surface [35].

To date, the only described *Tafilalichthys* jaw specimen is an anterior supragathal [30], which indicated that *Tafilalichthys* was durophagous, although not specialised to the same degree as the related *Bungartius* or *Mylostoma* [26]. The *Tafilalichthys* inferognathal investigated herein (PIMUZ A/I 4717 - Figure 2) suggests that *Tafilalichthys* may have been more adapted for durophagy than previously thought, with the anterior symphyseal region somewhat resembling that previously described for *Bungartius* and other durophagous arthrodires – with the occlusal dorsal surface partially composed of a cancellous texture [36]. However, this surface is flattened to the point of horizontality in *Tafilalichthys*, whereas both *Bungartius* [35] and *Mylostoma* [37] have more curved dental regions – which potentially could also have ‘chopped’ prey [38].

Like *Mylostoma*, the posterior ‘blade’ portion of *Tafilalichthys*’ inferognathal comprises over half of the total length, as opposed to a smaller proportion in the earlier, Frasnian (383-372 Ma) mylostomatids – which were less specialised for durophagy [39]. This proportional lengthening of the blade is thought to have increased the area of attachment for the adductor (jaw-closing) muscles, thereby increasing the bite force; crucial when specialising upon tough to digest, hard-shelled prey [40].

Dunkleosteus was selected as a comparison due to its well-documented status as an apex predator [41] and an arthrodire - indicating fairly close relatedness with Mylostomatidae [8]. Ideally, a *Dunkleosteus marsaisi* specimen could have been located, as it co-occurred with *Titanichthys termieri* in the Southern Maïder basin [42], however this did not prove possible. Instead, *Dunkleosteus terrelli*, known from the Cleveland Shale, was used. While *D. terrelli* was substantially larger than *D. marsaisi*, the skulls of the two species seem to have broadly similar shapes [9]. Given that all jaws in this study were scaled to the same length, using either species would be likely to yield similar results.

The inferognathal of *Dunkleosteus terrelli* is more clearly differentiated into blade and dental portions than the other arthrodires in this study. The dental portion is divided into an anterior fang-shaped cusp, presumably for puncturing flesh, and a posterior sharp blade which occluded with a parallel bladed surface on the supragathal [27]. This masticating, bladed surface is part of the dental portion of the inferognathal, separate from the edentulous posterior portion [34]. From a simple visual comparison, it appears much better-adapted for consuming large prey than *Titanichthys*.

Extant taxa

Sharks were selected as an extant comparison group due to the range of feeding strategies they display, including taxa with potentially analogous lifestyles to the three arthrodiran species investigated. The basking shark (*Cetorhinus maximus*) is a megaplanktivore, approaching a body length of 12 m [43]. Being closely related to an apex predator it co-occurs with, the great white shark (*Carcharodon carcharias*), the basking shark seems analogous with the proposed ecological niche of *Titanichthys*. The whale shark (*Rhincodon typus*) would potentially have represented an even closer analogue for *Titanichthys*, having also evolved from durophagous ancestors [44], as seems likely for *Titanichthys* – unfortunately whale shark specimens could not be accessed for this study.

The great white shark is an ideal analogue for *Dunkleosteus*, being a lamniform shark (the same order as *Cetorhinus* [45]) with a powerful bite force befitting of an apex predator [46]. The horn shark *Heterodontus francisci* was selected for its durophagous lifestyle [47], making it analogous for the proposed feeding strategy of *Tafilalichthys*. However, it is not that closely related to the other sharks in this study; being in a different order, the Heterodontiformes [48]. Due to the absence of known durophagous species among lamniform sharks, *Heterodontus* is the most suitable candidate for a durophage related to *Cetorhinus*.

To provide a further comparison point, and potentially assess whether certain lower jaw structural changes were common among parallel evolutionary pathways, whales were also included in the analysis. The planktivorous blue whale (*Balaenoptera musculus*) was compared with the killer whale (*Orcinus orca*), an apex predator [49]. A third comparison species was not used because of the lack of durophagous whale species. Due to the considerable evolutionary distance between the filter-feeding mysticetes and macrophagous odontocetes – which diverged around 38 Ma [50] – this comparison may be somewhat less strong. When the investigated species are co-occurring sister taxa, like *Titanichthys* and *Tafilalichthys*, morphological differences are more likely to be driven by a single explanatory factor, like divergence of function. There is a far greater possibility that differences between distantly-related species are due to a myriad of different factors, the effects of which are hard to distinguish between. Results for whales should be viewed with that caveat in mind.

Finite element model construction

All jaw models were produced using surface scans of the original specimens. Some specimens had already been scanned prior to this research (Table 1), those remaining were scanned at the University of Zurich using an Artec Eva light 3D scanner (Artec 3D). Surface scans were used instead of computerised tomography (CT) scans as the size and composition of some specimens rendered CT scanning extremely difficult. This, unfortunately, prevented the incorporation of internal features into the models; therefore, the jaws were treated as homogenous structures. Doing so has previously yielded differing results to more accurate, heterogenous models [46]. However, the surface scans should still prove valid for the purely shape-based comparison undertaken in this paper; although CT-scanning would be essential for an assessment of the absolute performance of *Titanichthys*' jaw. While the shark jaws were originally CT-scanned [51], only the surfaces were used to ensure methodological equivalence between species.

Jaw scans were processed, cleaned (removal of extraneous material and smoothing of fractures) and fused (where jaws were scanned in separate pieces) using a combination of Artec Studio 12 (Artec 3D), Avizo 9.4 (FEI Visualization Sciences Group) and MeshLab [52]. Jaw models were scaled to the same total length, as model size and forces applied had to be kept constant to ensure the analysis was solely investigating the effect of jaw shape on stress/strain resistance. Ideally, the models would have been scaled to the same surface area instead of length, as this typically produces stress comparisons of greater validity [53]. Similarly, scaling models to volume is most effective for comparing strain resistance. However, the extremely varied dentition among the various species skewed the results when models were scaled to either the same surface area or volume; an effect that has been noted previously [34]. Consequently, it was judged that equivocating model size using jaw length produced reasonably comparative models.

The muscle force applied to the jaws was adapted from a prior investigation of arthrodiran jaw mechanics [34] which primarily centred on a *Dunkleosteus terrelli* inferognathal. Consequently, all jaws were scaled to the length of the *D. terrelli* inferognathal scanned herein. The material properties proposed by Snively *et al.* [34], based on typical arthrodiran inferognathals, were applied to all jaw models. Treating each jaw as one homogenous material, jaws were assigned a Young's modulus of 20 GPa and a Poisson's ratio of 0.3. A vertical force of 300N was applied at the presumed central point of adductor mandibulae attachment. While this does

not accurately represent the force exerted by the muscles, it is a decent approximation given the absence of further skull material with which muscle action could be modelled [27]. Each jaw was constrained at the attachment point with the skull – typically on the dorsal surface at the posterior end of the lower jawbone. This constraint involved fixing a node at the attachment point for both translation and rotation in the X, Y and Z axes. Another constraint was applied to a node at the base of the anteriormost tooth (or the roughly analogous location proportionally for species with no discernible dentition), fixed for translation in the Y axis – effectively simulating the dentition being suspended within an item of prey.

Each jaw scan was ‘meshed’ – divided into elements, comprising the 3D volume of the jaw – in Hypermesh (Altair Hyperworks; Troy, Michigan), whereupon forces, constraints and material properties were applied. Each loaded model was imported into Abaqus (Dassault Systmes Simulia Corp., Providence), where FEA was performed. Every element is comprised from multiple nodes, which make up the outline of the element. Given the material properties of the model and the applied constraints, the deformation at each node can be simulated using FEA [29]. From these deformations, the stresses and strains experienced by each element within the model can be calculated.

The primary indicator selected was von Mises stress, which relates to the likelihood of ductile yielding causing a structure to fail [53]. Maximum principal stress distribution across the jaw was also analysed, as an indicator of the probability of brittle fracture. Given that bone responds in both ductile and brittle manners to stresses [54], recording both von Mises and maximum principal stress values should provide a more comprehensive profile of the jaws’ robustness. The maximum principal strain value of each element was also recorded. The extent of strain experienced within a structure indicates the degree of deformation undergone by the structure, therefore models with lower strain values are more resistant to deformation [53].

Experimentally, it was observed that proportional comparisons based on each of the three metrics produced extremely similar results (see Supplementary). Consequently, only von Mises stress was used for further analysis, as it seemed to reflect structural robustness effectively.

It is important to emphasise that the values displayed herein are very unlikely to accurately represent the actual values of stress the jaws would experience. Re-scaling of the jaw length, as well as assignment of equal material properties and applied forces, renders the absolute values irrelevant. Instead, these measures all served to validate comparisons between the different finite element models. Consequently, it is the proportional differences between the stress values experienced across the respective jaws that should be the main focus of analysis, as the disparities observed will indicate the relative robustness of the jaw shapes.

Finite element analysis

Initial comparison of stress distribution across the finite element models will be purely visual, which has been used repeatedly to effectively distinguish mandibles by their dietary function [34,55]. This will enable qualitative assessment of the stress patterns in the respective jaws, highlighting regions of particularly high stress and enabling an approximation of the differing overall resistances to stress.

In order to compare between the jaws quantitatively, average von Mises stress values were recorded for each model, from every element across the model. Typically, mean values are used [56], but median values may prove more robust to being skewed by extreme values [57]. Consequently, both mean and median values were calculated for *Titanichthys*, whereupon the value of the respective metrics could be assessed. Averaging has the advantage of enabling comparison of total stress and strain resistance with a far greater degree of precision than from pure visual comparison [58]. When combined with visual comparison, particularly weak or robust sections of the structure can still be identified. In addition, Kruskal-Wallis tests were used to assess for significance in any disparities between species’ median von Mises stress values [59] – although the massive sample sizes of the underlying data, with some models having over 200,000 elements, are likely to imbue even small differences with statistical significance.

However, averaging results can be skewed by element size, with smaller elements typically yielding more accurate results [60]. To combat this, an ‘Intervals method’ has been proposed [58], which incorporates element volumes to provide a more valid result. This method could allow for considerably more effective comparison of finite element models, and consequently more precise distinction between feeding strategies.

Intervals Method

The full method is described in the original paper [58], but will be outlined in brief here. Following FEA, all elements in the model are sorted by their von Mises stress value. These are then grouped into a number of

‘intervals’, each of which has an equal range of stress values. 50 intervals proved the optimal amount in the original experiment, so are used in this test (however as few as 15 intervals were still broadly effective at discriminating between dietary functions).

The cumulative volume of the elements represented in each of the 50 intervals can be calculated, then represented as a percentage of the total model volume. This represents the distribution of stresses across a model, characterising the proportion of the elements experiencing particular stress levels. A principal component analysis (PCA) is performed, based on the percentage of jaw volume represented in each interval, plotting the jaw models on two axes (principal components) that should describe the majority of variation in stress distribution. Species with similar diets should group together to an extent, if the differences in stress distribution between feeding strategies can be categorised. This method has successfully distinguished between different dietary preferences in jaw models previously, although using species within the same genus [58], much more closely related than the species tested here. Experimentally, it was observed that the whale jaw models were poorly-suited for direct comparison with the other species, as the considerable morphological disparity resulted in some models being represented in less than half of the stress intervals. Consequently, whales were removed from the PCA, to prevent skewing of the results.

4. Results

Visual comparison

The magnitudes of von Mises stress vary significantly between the placoderm inferognathals, but the general stress distribution patterns are relatively consistent (Figure 3). The highest von Mises stress values for all three species occur in the posterior bladed region; particularly close to the jaw attachment point, likely as a result of the constraint applied there. Higher stress values are experienced on the lateral aspects of each jaw than the medial. The fixed anterior point is also associated with high stress, but these regions are much more localised than at the jaw and muscle attachment points. *Titanichthys* exhibits the least resistance to von Mises stress among the placoderms, with *Dunkleosteus* proving the most resistant.

Visually, *Carcharodon* appears to be the least resistant to von Mises stress of the three shark lower jaws (Figure 3), with *Heterodontus* likely the most resistant. In whales, the mandible of *Orcinus* is clearly more resistant than *Balaenoptera*, which is characterised by extremely high levels of von Mises stress, experienced across the majority of the structure (Figure 3).

Quantitative comparison

Averaging the per element von Mises stress values produced differing results depending on whether the median or mean was used. However, while the actual values produced diverged (Table 2), the proportional differences between the species remained relatively consistent. Consequently, either method seems equally applicable; to simplify the results, the median will be used as the method of averaging henceforth.

Among the placoderms, the inferognathal of *Titanichthys* was the least resistant by some margin (Figure 4). The median elemental von Mises stress value for the inferognathal of *Tafilalichthys* represented 71% of the equivalent figure for *Titanichthys*, while in *Dunkleosteus* it was just 37%.

In general, the average von Mises stress values for shark jaws (Figure 4) were lower than in placoderms, with the highest value in sharks (in *Cetorhinus*) only slightly (0.1 MPa) higher than the lowest value in placoderms – for *Dunkleosteus*. While the jaws are typically more robust in sharks, there are some similar patterns when comparing proportional differences between the sharks. The filter-feeding basking shark displays the highest average stress, although the difference between it and the macropredatory great white shark is notably smaller than the (potentially) corresponding disparity between *Titanichthys* and *Dunkleosteus*. The median von Mises stress for *Carcharodon* is 75% of the equivalent for *Cetorhinus*, a disparity dwarfed by the much greater resistance to stress observed in *Heterodontus* (28% of *Cetorhinus*).

With a median von Mises stress value of 9.32 MPa, the mandible of *Balaenoptera musculus* is markedly less resistant to stress than all other jaws investigated (Figure 4). There is a large inter-lineage disparity with the von Mises stress resistance of *Orcinus*, the median of which is 19% of that of *Balaenoptera*.

Kruskal-Wallis tests revealed the differences between the median von Mises stress values for each species to be highly significant ($p < 0.0001$).

Intervals method

The intervals method PCA (Figure 5) attempts to differentiate between species based on the distribution of stress across the jaw models. The method groups *Titanichthys* with the planktivorous *Cetorhinus* and the closely related *Tafilalichthys*. There is little obvious diet-based grouping of the macrophagous species, with the non-*Cetorhinus* shark species relatively close together.

5. Discussion

Titanichthys' jaw conforms to a suspension feeding ecology

The inferognathal of *Titanichthys* was less resistant to von Mises stress than those of either *Dunkleosteus* or *Tafilalichthys*. *Dunkleosteus terrelli* has been substantively established as an apex predator [27,41], while *Tafilalichthys* can be considered to have been durophagous with some confidence, due to its morphological resemblance to, and close relatedness with, known durophagous arthrodires [8,26]. The comparatively high levels of stress observed in the inferognathal of *Titanichthys* suggest that neither feeding strategy would have been possible for *Titanichthys*, as its inferognathal would likely have failed (either by ductile yielding or brittle fracture) if exposed to the forces associated with the alternate feeding strategies. This strongly suggests that it was indeed a filter-feeder, as predicted based on its jaw morphology.

If *Titanichthys* were a filter-feeder, the primary function of its jaw would have been to maximise the water taken into the oral cavity during feeding, thereby increasing the rate of prey intake [61]. Morphologically, the inferognathal of *Titanichthys* seems ideally suited for this purpose – its elongation increased the maximum capacity of the oral cavity, which correlates with water filtration rate [62]. The perceived elongation of *Titanichthys*' inferognathal can be demonstrated by comparing its size with an inferognathal of the similarly-sized *Dunkleosteus terrelli*, which is clearly wider and shorter - the specimen used here is less than half the length of the corresponding *Titanichthys* inferognathal [63]. The narrowing of *Titanichthys*' inferognathal, associated with elongation, would have reduced its mechanical robustness (as displayed in this study). This adaptation would likely be unfeasible for a species reliant upon consuming large or hard-shelled prey, as it would result in a fitness reduction from an adaptive peak [64]. Should planktivory have been possible for *Titanichthys*, it could have been freed from evolutionary constraints preventing any weakening of its inferognathal; replaced by selection for maximising prey intake.

While the inferognathals of both species were considerably more mechanically resilient than that of *Titanichthys*, there is still a sizable disparity between the von Mises stress values observed in *Dunkleosteus terrelli* and *Tafilalichthys lavocati*. Biomechanical analyses have suggested that *Dunkleosteus* was capable of feeding on both highly-mobile and armoured prey [27], due to its high bite force and rapid jaw kinematics. In rodents, species with generalist diets have been shown to be more resistant to stresses across the skull than their more specialist relatives [65]. It is possible the comparatively generalist *Dunkleosteus* had a more stress-resistant inferognathal than the specialist durophage *Tafilalichthys* for the same reason.

In sharks, the highest values of stress are seen in the filter-feeding basking shark. This adds weight to the conclusion that *Titanichthys* was a filter-feeder, as the obligate planktivorous shark is significantly less resistant to stress than its durophagous and macropredatory relatives. The disparity in stress resistance between the lower jaws of *Carcharodon* and *Cetorhinus* is smaller than the equivalent disparity between *Titanichthys* and *Dunkleosteus*. The basking shark's lower jaw retains the same basic structure, albeit with less complexity, of the other shark species; whereas the lower jaw of *Titanichthys* is more morphologically divergent from the other placoderm species investigated, probably causing the more disparate results.

It is notable that, while there is a large difference in lower jaw robustness between *Cetorhinus* and *Titanichthys* (median von Mises stress of 0.78 MPa compared with 1.83 MPa, respectively), *Carcharodon* and *Dunkleosteus* performed very similarly. The median von Mises stress for *Carcharodon* was 85% of the respective value for *Dunkleosteus*. *Dunkleosteus* likely occupied the equivalent niche as *Carcharodon*, but their methods of subduing prey probably differed as a result of very efficient locomotion in the great white shark [66], which is unlikely to have been replicated in the heavy, less streamlined *Dunkleosteus* [67]. Consequently, the great white shark lower jaw was expected to prove more resistant to stress than the inferognathal of *Dunkleosteus* – and this may have been seen to a greater extent if cartilaginous properties were applied to the shark. Treating a great white shark jaw as homogenous bone has previously resulted in underestimated stress resistance [46] and a lower Young's modulus associated with calcified cartilage would result in higher jaw strain. On the other hand,

prior research indicating that the bite force: body mass ratio of *Dunkleosteus* is roughly equivalent to that of the great white shark [27] suggests that similar stress resistances, when scaled to length, are to be expected. The stress resistance of the great white shark's lower jaw may have been roughly equivalent to that of *Dunkleosteus*, but no such resemblance between potential analogues was observed in the durophagous species. The lower jaw of *Heterodontus francisci* is a thick structure devoid of ornamentation beyond its dentition, which proved to be substantially more robust than the lower jaw of any other species investigated. The mass-specific bite force of *Heterodontus francisci* has been shown to markedly exceed that of *Carcharodon carcharias* [68], enabling efficient crushing of its hard-shelled prey. Consequently, the disparity in stress resistance between the two species is not unexpected.

What initially seems more surprising is the even larger difference between the jaw robustness of the two durophagous species, with the horn shark being far more resilient than *Tafilalichthys*. Their roughly equivalent diets would suggest similar mechanical requirements of their jaws; however, the disparity may be explained by behavioural differences. Durophagous placoderms are thought to have primarily broken down the hard shells of their prey utilising shearing, as opposed to the more mechanically taxing, crushing mechanism seen in chondrichthyans and other post-Devonian fish [69]. This suggests that the jaws of *Tafilalichthys* would have experienced less stress than those of the shell-crushing *Heterodontus* [47].

It is worth noting that the shark finite element models were produced using surface scans originally created for use in a geometric morphometric study [51]. Consequently, they were not ideally suited to being discretised into a single, uniform surface. Despite extensive remeshing using both Blender [70] and Hypermesh, the shark jaw models were still of poorer quality than the other jaw models. The impact of this on the overall results are difficult to determine, but it should be kept in mind that the broad patterns are of more use than any specific numerical values.

The fundamental pattern outlined within this study is demonstrated further in whales: the mandible of the filter-feeding blue whale is less resistant to von Mises stress than that of the macropredatory killer whale, but with a far greater disparity than in the other lineages. Jaw elongation is seen to a far greater degree in the mysticete whales than the other megaplanktivorous lineages investigated, probably as a consequence of the energetically-expensive 'lunge feeding' method utilised by most mysticetes [71]. This is doubly true for the massive jaws of the blue whale, which enable incredibly efficient feeding despite substantial mechanical expenditure [72]. Consequently, resistance to stress may be lowest in the blue whale jaw as a result of maximising feeding efficiency.

The mandible of *Orcinus* is considerably more resistant to stress than that of *Balaenoptera*, but the median values are still notably higher than in *Carcharodon* and *Dunkleosteus*, the proposed analogues of the killer whale. Ecological reasons for this are difficult to determine, with the typical diet of an orca resembling the diet of a great white shark: centering on marine mammals [73] but sufficiently generalist to predate a wide range of species [74]. This ecological similarity would seem to suggest roughly equivalent jaw robustness, a pattern, which is not seen here.

Methodological factors may have impacted the modelling results for the whale jaws. The orca's teeth were not attached to the scanned mandible. Teeth were generally associated with relatively low stress values in this study, removing these regions from the model may have raised the average values. Manually attaching them to the digital model was considered, but the imperfect nature of this would probably have further reduced the validity of the model; similarly, removing dentition of basking shark jaws or the bone parts used for cutting or crushing in placoderm jaws would have been impossible.

All jaw scans were scaled to the same length, to circumvent the impact of teeth on scaling to the same surface area. While this seemed to improve the validity of comparisons between the model placoderm and shark jaws, it may have had the opposite effect with the whale jaws. Scaling the blue whale jaw rendered it extremely narrow relative to the other jaws, to an unrealistic extent. This may partially explain the average stress value calculated in the blue whale jaw massively exceeding those of any other species. Indeed, when the whale jaws were scaled to the same surface area, the average stress values in the orca's jaw were around 70% of the equivalent values in the blue whale. By contrast, re-scaling had little impact on the orca's jaw robustness compared with the other apex predators. Again, the large-scale trends are much more valuable than any specific numerical values – and using either method revealed that the megaplanktivore jaw was significantly less mechanically resilient than that of the apex predator.

The intervals method is probably better-suited to comparing between more closely-related species [58], as their morphology would likely be more homogenous – making function-related divergences more central to

the analysis. Despite the vast evolutionary distances involved, the method effectively grouped the planktivorous *Cetorhinus* together with *Titanichthys*. This may suggest that *Titanichthys* was a filter-feeder, as the jaws of *Titanichthys* and *Cetorhinus* are very distinct morphologically, yet consistent patterns in von Mises stress distribution between them are statistically quantifiable. The close placement of *Titanichthys* and *Tafilalichthys* does suggest some caution should be taken with any interpretation, although this is likely more a function of their close relatedness than of a shared ecological niche. The PCA did not group the durophagous or macropredatory species together, although this was predictable to an extent as the stress values of those species seemed to be more influenced by their lineage than their diet. Despite this, the intervals method's detection of corresponding stress patterns between (potentially) filter-feeding species is notable. This method should be applied in a variety of contexts moving forward, to assess for mechanical adaptations underlying functional divergences in other lineages.

Trajectories in megaplanktivory – durophagous origins?

Complete *Tafilalichthys* inferognathals are not previously figured in the literature. Consequently, the specimen described in this study is significant for advancing our understanding of arthrodiran interrelationships and the evolutionary pathway that seemingly resulted in obligate planktivory in *Titanichthys*. The morphology and mechanical performance of the inferognathal both indicate that durophagy was the most likely feeding strategy for *Tafilalichthys*, supporting its proposed phylogenetic position within the Mylostomatidae [8]. With all the major mylostomatids (excluding *Titanichthys*) likely to have been durophagous, it seems reasonable to conclude that *Titanichthys* evolved from durophagous ancestors.

Evolutionary transitions from durophagy to planktivory have occurred a number of times. The filter-feeding whale shark (*Rhincodon typus*) arose from the typically benthic Orectolobiformes [44]. Its closest relative, the nurse shark (*Ginglymostoma cirratum*), is durophagous, feeding principally on hard-shelled invertebrates [75]. Similarly the sister taxon of the planktivorous Mobulidae (manta and devil rays) are the durophagous Rhinoptera (cownose rays) [76,77]. In a less clear parallel, the only pinniped proposed to have been durophagous [78] was relatively closely related to the ancestor of the Lobodontini- pinnipeds uniquely specialised for planktivory [79].

Temporal evolution and extinction of megasuspension feeders

The emergence of a megaplanktivore in the Famennian may hold similar clues to the degree and nature of marine primary productivity during the Devonian period [80]. Modern forms migrate to regions of high seasonal productivity, such as mysticetes seeking arctic oceans and highly productive upwelling zones. Basking sharks focus on relatively less productive seasonal blooms in shallow boreal and warm temperate waters, while whale sharks are associated to tropical waters and seasonal blooms and spawning events in this realm. The evolution of megaplanktivores coincided with periods of high productivity [15,16]. For example, the radiation of mysticete whales coincide with the Neogene cooling pump and onset of the circumantarctic polar current, resulting in a stronger thermohaline pump. It has been noted that the emergence of suspension feeding pachycormid fishes correlate with the evolution of key phytoplankton: Dinoflagellates, diatoms and coccolithosphorids could reflect the increase in primary productivity that led to the Mesozoic Marine revolution [81]. While perhaps not necessarily being drivers of the revolution, the conditions permissive of such a radiation in marine primary producers may indeed reflect a marked shift in opportunity. Similarly, suspension feeding radiodonts during the Cambrian explosion [4,82,83] radiated synchronously with the first establishment of a tiered food chain with several (at least four) levels of consumers. While the Cambrian radiation may be entirely unique with the innovation of micropredation [84], evidence for increased primary productivity is manifested in global early Cambrian phosphate deposits [85] often associated to upwelling systems in modern oceans [86].

The Devonian saw the first emergence of arborescent plants on land [23]. This resulted in deeper rooting systems, higher silicate rock weathering and nutrient run off into the oceans. While increasing primary productivity, it also led to near global deep ocean anoxia, black shale deposition [87,88] and the Frasnian-Famennian Kellwasser event, one of the 'big five' mass extinctions [89]. The increased nutrients going into circulation may well have been the necessary push for allowing arthrodires to explore this ecological niche of megaplanktivory as the first vertebrates on record.

The apparent punctuation and compelling correlation between major marine radiations, shifts in apparent productivity and megaplanktivores may be of interest for understanding how this unique ecological strategy

respond to global perturbations, such as human induced climate change. With their potential added sensitivity, megaplanktivores may be 'canary birds' for ocean ecosystem health. Some caution is advised, however. There may be taphonomic biases preventing the recognition of each and every megaplanktivore in existence at a given time. As with *Titanichthys*, tell-tale features of ecology may have been lost during fossilisation. As a rule one would want to have the filter feeding apparatus preserved, but otherwise other associated anatomical adaptations or stomach contents will need to be identified [90]. There are almost certainly other planktivorous species in the fossil record yet to be identified, shown by the recent re-appraisal of the Cretaceous plesiosaur *Morturneria seymourensis* as a probable filter-feeder [91]. Indeed, there are even other placoderms that may have been planktivorous: the arthrodire *Homostius* had a narrow jaw devoid of dentition or shearing surfaces and substantially pre-dated *Titanichthys* [92]. The common reduction in stress/strain resistance observed here could be used as an indicator of planktivory in such cases where it seems plausible but cannot be identified definitively, due to the absence of fossilised filtering structures.

6. Conclusion

Finite element analysis of the lower jaw of *Titanichthys* revealed that it was significantly less resistant to von Mises stress than those of related arthrodiras that utilised macrophagous feeding strategies. This suggests that these strategies would not have been viable for *Titanichthys*, as its jaw would have been insufficiently mechanically robust. Consequently, it is highly likely that *Titanichthys* was a filter-feeder – a feeding method that is likely to exert considerably less stress on the jaw than macrophagous feeding modes. The validity of assigning filter-feeding based on jaw mechanical resilience is supported by the roughly equivalent patterns known from lineages containing extant filter-feeders.

Common morphological trends in the convergent evolution of megaplanktivores can be not only observed but quantified mechanically utilising finite element analysis. A variety of methods were used to compare between the jaw models, due to imperfections with solely comparing visually or using average stress. The intervals method grouped feeding strategies to an extent, providing an additional perspective.

Tafilalichthys, a member of the Mylostomatidae and probably one of *Titanichthys*' closest relatives, appears to have been durophagous. Durophagy is the likely feeding mode of all crown-group mylostomatid, except *Titanichthys*, suggesting that it evolved from a durophagous ancestor. This durophage-to-planktivore transition is surprisingly common among convergently-evolved giant filter-feeders: also seen in multiple, independently-evolved planktivorous elasmobranch lineages.

The presence of a megaplanktivore in the Famennian supports the theory that productivity was high in the Late Devonian, likely a result of increased eutrophication caused by the diversification of terrestrial tracheophytes and the advent of arborescence. It reflects the link between the increasing complexity of Devonian marine ecosystems and functional diversity of Arthrodira, which occupied a wide range of ecological niches. Most significantly, it reveals that vertebrate megaplanktivores likely existed over 150 Ma prior to the Mesozoic pachycormids, previously considered the earliest definitive giant filter-feeders.

Acknowledgments

We would like to thank Jordi Marcé-Nogué, for allowing us to utilise his technique and adjusting it for our data. Jaw scans were provided by the Natural History Museum in London, the Idaho Museum of Natural History and Matt Friedman; others were acquired from Pepijn Kamminga's online repository of shark jaw models [49] – we extend our gratitude to them all. We also thank James Boyle for his insight and translation of relevant literature.

Ethical Statement

This article does not present research with ethical considerations. No live animals were used in the study.

Funding Statement

CK was supported by the Swiss National Science Foundation SNF (project nr. 200020_184894).

Data Accessibility

The datasets supporting this article have been uploaded as part of the Supplementary Material.

Competing Interests

We declare that we have no competing interests.

Authors' Contributions

S.J.C, E.J.R and J.V designed the study. C.K provided and scanned the Moroccan placoderm specimens. S.J.C carried out all analysis and wrote the manuscript, with critical revision from all authors. All the authors gave final approval for submission.

References

- Sanderson SL, Wassersug R. 1990 Suspension-Feeding Vertebrates. *Sci. Am.* **262**, 96–102. (doi:10.2307/24996794)
- Costa DP. 2009 Energetics. *Encycl. Mar. Mamm.*, 383–391. (doi:10.1016/B978-0-12-373553-9.00091-2)
- Clapham P. 2001 Why do Baleen Whales Migrate?. *Mar. Mammal Sci.* **17**, 432–436. (doi:10.1111/j.1748-7692.2001.tb01289.x)
- Vinther J, Stein M, Longrich NR, Harper DAT. 2014 A suspension-feeding anomalocarid from the Early Cambrian. *Nature* **507**, 496–499. (doi:10.1038/nature13010)
- Friedman M, Shimada K, Martin LD, Everhart MJ, Liston J, Maltese A, Triebold M. 2010 100-million-year dynasty of giant planktivorous bony fishes in the Mesozoic seas. *Science* **327**, 990–3. (doi:10.1126/science.1184743)
- Carr RK. 1995 Placoderm diversity and evolution. *Bull. du Muséum Natl. d'Histoire Nat. 4ème série – Sect. C – Sci. la Terre, Paléontologie, Géologie, Minéralogie* **17**, 85–125.
- Percival LME, Davies JHFL, Schaltegger U, De Vleeschouwer D, Da Silva A-C, Föllmi KB. 2018 Precisely dating the Frasnian–Famennian boundary: implications for the cause of the Late Devonian mass extinction. *Sci. Rep.* **8**, 9578. (doi:10.1038/s41598-018-27847-7)
- Boyle J, Ryan MJ. 2017 New information on *Titanichthys* (Placodermi, Arthrodira) from the Cleveland Shale (Upper Devonian) of Ohio, USA. *J. Paleontol.* **91**, 318–336. (doi:10.1017/jpa.2016.136)
- Denison RH. 1978 *Placodermi*. Volume 2. München.
- Sanderson SL, Wassersug R. 1993 Convergent and alternative designs for vertebrate suspension feeding. In *The Skull* (eds J Hanken, BK Hall), pp. 37–112. University of Chicago Press.
- Schumacher BA, Shimada K, Liston J, Maltese A. 2016 Highly specialized suspension-feeding bony fish *Rhinconichthys* (Actinopterygii: Pachycormiformes) from the mid-Cretaceous of the United States, England, and Japan. *Cretac. Res.* **61**, 71–85. (doi:10.1016/J.CRETRES.2015.12.017)
- Liston J. 2013 The plasticity of gill raker characteristics in suspension feeders: Implications for Pachycormiformes. In *Mesozoic Fishes 5 - Global Diversity and Evolution* (eds G Arratia, HP Schultze, MVH Wilson), pp. 121–143. München: Verlag Dr. Friedrich Pfeil.
- Brazeau MD, Friedman M, Jerve A, Atwood RC. 2017 A three-dimensional placoderm (stem-group gnathostome) pharyngeal skeleton and its implications for primitive gnathostome pharyngeal architecture. *J. Morphol.* **278**, 1220–1228. (doi:10.1002/jmor.20706)
- Berger WH. 2007 Cenozoic cooling, Antarctic nutrient pump, and the evolution of whales. *Deep Sea Res. Part II Top. Stud. Oceanogr.* **54**, 2399–2421. (doi:10.1016/J.DSR2.2007.07.024)
- Marx FG, Uhen MD. 2010 Climate, critters, and cetaceans: Cenozoic drivers of the evolution of modern whales. *Science* **327**, 993–6. (doi:10.1126/science.1185581)
- Pimiento C, Cantalapiedra JL, Shimada K, Field DJ, Smaers JB. 2019 Evolutionary pathways toward gigantism in sharks and rays. *Evolution (N. Y.)* **73**, 588–599. (doi:10.1111/evo.13680)
- Álvaro JJ, Ahlberg P, Axheimer N. 2010 Skeletal carbonate productivity and phosphogenesis at the lower–middle Cambrian transition of Scania, southern Sweden. *Geol. Mag.* **147**, 59. (doi:10.1017/S0016756809990021)
- Martin RE. 1996 Secular Increase in Nutrient Levels through the Phanerozoic: Implications for Productivity, Biomass, and Diversity of the Marine Biosphere. *Palaios* **11**, 209. (doi:10.2307/3515230)
- Niklas KJ, Tiffney BH, Knoll AH. 1983 Patterns in vascular land plant diversification. *Nature* **303**, 614–616. (doi:10.1038/303614a0)
- Morris JL *et al.* 2015 Investigating Devonian trees as geo-engineers of past climates: linking palaeosols to palaeobotany and experimental geobiology. *Palaeontology* **58**, 787–801. (doi:10.1111/pala.12185)
- Berner RA. 1997 The Rise of Plants and Their Effect on Weathering and Atmospheric CO₂. *Science (80-)* **276**, 544–546. (doi:10.1126/science.271.5252.1105)
- Le Hir G, Donnadiou Y, Goddérés Y, Meyer-Berthaud B, Ramstein G, Blakey RC. 2011 The climate

- change caused by the land plant invasion in the Devonian. *Earth Planet. Sci. Lett.* **310**, 203–212. (doi:10.1016/j.epsl.2011.08.042)
23. Algeo TJ, Scheckler SE. 1998 Terrestrial-marine teleconnections in the Devonian: links between the evolution of land plants, weathering processes, and marine anoxic events. *Philos. Trans. R. Soc. B Biol. Sci.* **353**, 113–130. (doi:10.1098/rstb.1998.0195)
24. Bambach RK. 1999 Energetics in the global marinefauna: A connection between terrestrial diversification and change in the marine biosphere. *Geobios* **32**, 131–144. (doi:10.1016/S0016-6995(99)80025-4)
25. Fletcher TM, Janis CM, Rayfield EJ. 2010 Finite Element Analysis of Ungulate Jaws: Can mode of digestive physiology be determined? *Palaeontol. Electron.* **13**, 15.
26. Lelièvre H. 1991 New information on the structure and the systematic position of *Tafilalichthys lavocati* (placoderm, arthrodire) from the Late Devonian of Tafilalet, Morocco. In *Early vertebrates and related problems of evolutionary biology* (eds M Chang, Y Liu, G Zhang), pp. 121–130.
27. Anderson PSL, Westneat MW. 2009 A biomechanical model of feeding kinematics for *Dunkleosteus terrelli* (Arthrodira, Placodermi). *Paleobiology* **35**, 251–269. (doi:10.1666/08011.1)
28. Rayfield EJ. 2007 Finite Element Analysis and Understanding the Biomechanics and Evolution of Living and Fossil Organisms. *Annu. Rev. Earth Planet. Sci.* **35**, 541–576. (doi:10.1146/annurev.earth.35.031306.140104)
29. Bright JA. 2014 A review of paleontological finite element models and their validity. *J. Paleontol.* **88**, 760–769. (doi:10.1666/13-090)
30. Lehman JP. 1956 Les Arthrodières du dévonien supérieur du Tafilalet:(Sud marocain). *Éditions du Serv. géologique du Maroc* **129**.
31. Derycke C, Olive S, Groessens E, Goujet D. 2014 Paleogeographical and paleoecological constraints on paleozoic vertebrates (chondrichthyans and placoderms) in the Ardenne Massif: Shark radiations in the Famennian on both sides of the Palaeotethys. *Palaeogeogr. Palaeoclimatol. Palaeoecol.* **414**, 61–67. (doi:10.1016/j.palaeo.2014.07.012)
32. Frey L, Pohle A, Rücklin M, Klug C. 2019 Fossil-Lagerstätten, palaeoecology and preservation of invertebrates and vertebrates from the Devonian in the eastern Anti-Atlas, Morocco. *Lethaia*, let.12354. (doi:10.1111/let.12354)
33. Anderson PSL. 2008 Shape variation between arthrodire morphotypes indicates possible feeding niches. *J. Vertebr. Paleontol.* **28**, 961–969. (doi:10.1671/0272-4634-28.4.961)
34. Snively E, Anderson PSL, Ryan MJ. 2010 Functional and ontogenetic implications of bite stress in arthrodire placoderms. *Kirtlandia* **57**, 53–60.
35. Dunkle DH. 1947 A new genus and species of arthrodire fish from the Upper Devonian Cleveland Shale. *Sci. Publ. Clevel. Museum Nat. Hist.* **8**, 103–117.
36. Young GC. 2004 A homostiid arthrodire (placoderm fish) from the Early Devonian of the Burrinjuck area, New South Wales. *Alcheringa An Australas. J. Paleontol.* **28**, 129–146. (doi:10.1080/03115510408619278)
37. Dunkle DH, Bungart PA. 1945 A new arthrodire fish from the Upper Devonian Ohio shales. *Sci. Publ. Clevel. Museum Nat. Hist.* **8**, 85–95.
38. Anderson PSL. 2009 Biomechanics, functional patterns, and disparity in Late Devonian arthrodires. *Paleobiology* **35**, 321–342. (doi:10.1666/0094-8373-35.3.321)
39. Hlavín WJ, Boreške JR. 1973 *Mylostoma variabile* Newberry, an Upper Devonian durophagous brachyothoracid arthrodire, with notes on related taxa. *Breviora* **412**.
40. Dunkle DH, Bungart PA. 1943 Comments on *Diplognathus mirabilis* Newberry. *Sci. Publ. Clevel. Museum Nat. Hist.* **8**, 73–84.
41. Anderson PS., Westneat MW. 2007 Feeding mechanics and bite force modelling of the skull of *Dunkleosteus terrelli*, an ancient apex predator. *Biol. Lett.* **3**, 77–80. (doi:10.1098/rsbl.2006.0569)
42. Cloutier R, Lelièvre H. 1998 Comparative study of the fossiliferous sites of the Devonian. *Prep. ministère l'Environnement la Faune, Gouv. du Québec*
43. Sims DW. 2008 Chapter 3 Sieving a Living: A Review of the Biology, Ecology and Conservation Status of the Plankton-Feeding Basking Shark *Cetorhinus Maximus*. *Adv. Mar. Biol.* **54**, 171–220. (doi:10.1016/S0065-2881(08)00003-5)
44. Goto T. 2001 Comparative Anatomy, Phylogeny and Cladistic Classification of the Order Orectolobiformes (Chondrichthyes, Elasmobranchii). *Mem. Grad. Sch. Fish. Sci. Hokkaido Univ.* **48**, 1–100.
45. SHIMADA K. 2005 Phylogeny of lamniform sharks (Chondrichthyes: Elasmobranchii) and the contribution of dental characters to lamniform systematics. *Paleontol. Res.* **9**, 55–72. (doi:10.2517/prpsj.9.55)
46. Wroe S *et al.* 2008 Three-dimensional computer analysis of white shark jaw mechanics: how hard can a great white bite? *J. Zool.* **276**, 336–342. (doi:10.1111/j.1469-7998.2008.00494.x)
47. Huber DR, Eason TG, Hueter RE, Motta PJ. 2005 Analysis of the bite force and mechanical design of the feeding mechanism of the durophagous horn shark *Heterodontus francisci*. *J. Exp. Biol.* **208**, 3553–71. (doi:10.1242/jeb.01816)
48. Vélez-Zuazo X, Agnarsson I. 2011 Shark tales: A molecular species-level phylogeny of sharks (Selachimorpha, Chondrichthyes). *Mol. Phylogenet. Evol.* **58**, 207–217. (doi:10.1016/J.YMPREV.2010.11.018)
49. Ford JKB, Ellis GM, Olesiuk PF, Balcomb KC. 2010 Linking killer whale survival and prey abundance: food limitation in the oceans' apex predator? *Biol. Lett.* **6**, 139–42. (doi:10.1098/rsbl.2009.0468)
50. Marx FG, Fordyce RE. 2015 Baleen boom and bust: a synthesis of mysticete phylogeny, diversity and disparity. *R. Soc. Open Sci.* **2**, 140434–140434. (doi:10.1098/rsos.140434)
51. Kamminga P, De Bruin PW, Geleijns J, Brazeau MD. 2017 X-ray computed tomography library of shark anatomy and lower jaw surface models. *Sci. Data* **4**, 170047. (doi:10.1038/sdata.2017.47)
52. Cignoni P, Callieri M, Corsini M, Dellepiane M, Ganovelli F, Ranzuglia G. 2008 MeshLab: an Open-Source Mesh Processing Tool.
53. Dumont ER, Grosse IR, Slater GJ. 2009 Requirements for comparing the performance of finite element models of biological structures. *J. Theor. Biol.* **256**, 96–103. (doi:10.1016/J.JTBI.2008.08.017)
54. Shigemitsu R, Yoda N, Ogawa T, Kawata T, Gunji Y, Yamakawa Y, Ikeda K, Sasaki K. 2014 Biological-data-based finite-element stress analysis of mandibular bone with implant-supported overdenture. *Comput. Biol. Med.* **54**, 44–52. (doi:10.1016/J.COMPBIOMED.2014.08.018)
55. Serrano-Fochs S, De Esteban-Trivigno S, Marcé-Nogué J, Fortuny J, Fariña RA. 2015 Finite Element Analysis of the Cingulata

- Jaw: An Ecomorphological Approach to Armadillo's Diets. *PLoS One* **10**, e0120653. (doi:10.1371/journal.pone.0120653)
56. Lautenschlager S. 2017 Functional niche partitioning in Therizinosauria provides new insights into the evolution of theropod herbivory. *Palaeontology* **60**, 375–387. (doi:10.1111/pala.12289)
57. Dar FH, Meakin JR, Aspden RM. 2002 Statistical methods in finite element analysis. *J. Biomech.* **35**, 1155–1161. (doi:10.1016/S0021-9290(02)00085-4)
58. Marcé-Nogué J, De Esteban-Trivigno S, Püschel TA, Fortuny J. 2017 The intervals method: a new approach to analyse finite element outputs using multivariate statistics. *PeerJ* **5**, e3793. (doi:10.7717/peerj.3793)
59. Erhart P, Hylhik-Dürr A, Geisbüsch P, Kotelis D, Müller-Eschner M, Gasser TC, von Tengg-Kobligk H, Böckler D. 2015 Finite Element Analysis in Asymptomatic, Symptomatic, and Ruptured Abdominal Aortic Aneurysms: In Search of New Rupture Risk Predictors. *Eur. J. Vasc. Endovasc. Surg.* **49**, 239–245. (doi:10.1016/J.EJVS.2014.11.010)
60. Bright JA, Rayfield EJ. 2011 The Response of Cranial Biomechanical Finite Element Models to Variations in Mesh Density. *Anat. Rec. Adv. Integr. Anat. Evol. Biol.* **294**, 610–620. (doi:10.1002/ar.21358)
61. Sims DW. 1999 Threshold foraging behaviour of basking sharks on zooplankton: life on an energetic knife-edge? *Proc. R. Soc. B Biol. Sci.* **266**, 1437–1443. (doi:10.1098/rspb.1999.0798)
62. Goldbogen JA, Potvin J, Shadwick RE. 2010 Skull and buccal cavity allometry increase mass-specific engulfment capacity in fin whales. *Proceedings. Biol. Sci.* **277**, 861–8. (doi:10.1098/rspb.2009.1680)
63. Anderson PSL, Friedman M, Brazeau MD, Rayfield EJ. 2011 Initial radiation of jaws demonstrated stability despite faunal and environmental change. *Nature* **476**, 206–209. (doi:10.1038/nature10207)
64. Hansen TF. 2012 Adaptive Landscapes and Macroevolutionary Dynamics. In *The adaptive landscape in evolutionary biology* (eds EI Svensson, R Calsbeek), pp. 205–226. Oxford University Press.
65. Cox PG, Rayfield EJ, Fagan MJ, Herrel A, Pataky TC, Jeffery N. 2012 Functional Evolution of the Feeding System in Rodents. *PLoS One* **7**, e36299. (doi:10.1371/journal.pone.0036299)
66. Donley JM, Sepulveda CA, Konstantinidis P, Gemballa S, Shadwick RE. 2004 Convergent evolution in mechanical design of lamnid sharks and tunas. *Nature* **429**, 61–65. (doi:10.1038/nature02435)
67. Carr RK. 2010 Paleoeology of *Dunkleosteus terrelli* (Placodermi: Arthrodira). *Kirtlandia, Clevel. Museum Nat. Hist.* **57**, 36–55.
68. Kolmann MA, Huber DR, Motta PJ, Grubbs RD. 2015 Feeding biomechanics of the cownose ray, *Rhinoptera bonasus*, over ontogeny. *J. Anat.* **227**, 341–351. (doi:10.1111/joa.12342)
69. Brett CE. 2003 Durophagous Predation in Paleozoic Marine Benthic Assemblages. In *Predator—Prey Interactions in the Fossil Record* (eds PH Kelley, M Kowalewski, TA Hansen), pp. 401–432. Boston, MA: Springer US. (doi:10.1007/978-1-4615-0161-9_18)
70. Zoppè M, Porozov Y, Andrei R, Cianchetta S, Zini MF, Loni T, Caudai C, Callieri M. 2008 Using Blender for molecular animation and scientific representation.
71. Goldbogen JA *et al.* 2012 Scaling of lunge-feeding performance in rorqual whales: mass-specific energy expenditure increases with body size and progressively limits diving capacity. *Funct. Ecol.* **26**, 216–226. (doi:10.1111/j.1365-2435.2011.01905.x)
72. Goldbogen JA, Calambokidis J, Oleson E, Potvin J, Pyenson ND, Schorr G, Shadwick RE. 2011 Mechanics, hydrodynamics and energetics of blue whale lunge feeding: efficiency dependence on krill density. *J. Exp. Biol.* **214**, 131–46. (doi:10.1242/jeb.048157)
73. Ford J, Ellis G. 2006 Selective foraging by fish-eating killer whales *Orcinus orca* in British Columbia. *Mar. Ecol. Prog. Ser.* **316**, 185–199. (doi:10.3354/meps316185)
74. Ford JKB. 2009 Killer Whale: *Orcinus orca*. In *Encyclopedia of Marine Mammals* (eds WF Perrin, B Würsig, JGM Thewissen), pp. 650–657. Academic Press. (doi:10.1016/B978-0-12-373553-9.00150-4)
75. Matott MP, Motta PJ, Hueter RE. 2005 Modulation in Feeding Kinematics and Motor Pattern of the Nurse Shark *Ginglymostoma cirratum*. *Environ. Biol. Fishes* **74**, 163–174. (doi:10.1007/s10641-005-7435-3)
76. Collins AB, Heupel MR, Hueter RE, Motta PJ. 2007 Hard prey specialists or opportunistic generalists? An examination of the diet of the cownose ray, *Rhinoptera bonasus*. *Mar. Freshw. Res.* **58**, 135. (doi:10.1071/MF05227)
77. Dunn KA, McEachran JD, Honeycutt RL. 2003 Molecular phylogenetics of myliobatiform fishes (Chondrichthyes: Myliobatiformes), with comments on the effects of missing data on parsimony and likelihood. *Mol. Phylogenet. Evol.* **27**, 259–270. (doi:10.1016/S1055-7903(02)00442-6)
78. Amson E, de Muizon C. 2014 A new durophagous phocid (Mammalia: Carnivora) from the late Neogene of Peru and considerations on monachine seals phylogeny. *J. Syst. Palaeontol.* **12**, 523–548. (doi:10.1080/14772019.2013.799610)
79. Koretsky IA, Rahmat S., Peters N. 2014 Remarks on Correlations and Implications of the Mandibular Structure and Diet in Some Seals (Mammalia, Phocidae). *Vestn. Zool.* **48**, 255–268. (doi:https://doi.org/10.2478/vzoo-2014-0029)
80. Klug C, Kröger B, Kiessling W, Mullins GL, Servais T, Frýda J, Korn D, Turner S. 2010 The Devonian nekton revolution. *Lethaia* **43**, 465–477. (doi:10.1111/j.1502-3931.2009.00206.x)
81. Knoll AH, Follows MJ. 2016 A bottom-up perspective on ecosystem change in Mesozoic oceans. *Proc. R. Soc. B Biol. Sci.* **283**. (doi:10.1098/rspb.2016.1755)
82. Van Roy P, Daley AC, Briggs DEG. 2015 Anomalocaridid trunk limb homology revealed by a giant filter-feeder with paired flaps. *Nature* **522**, 77–80. (doi:10.1038/nature14256)
83. Lerosey-Aubril R, Pates S. 2018 New suspension-feeding radiodont suggests evolution of microplanktivory in Cambrian macronekton. *Nat. Commun.* **9**. (doi:10.1038/s41467-018-06229-7)
84. Sperling EA, Frieder CA, Raman A V., Girguis PR, Levin LA, Knoll AH. 2013 Oxygen, ecology, and the Cambrian radiation of animals. *Proc. Natl. Acad. Sci. U. S. A.* **110**, 13446–13451. (doi:10.1073/pnas.1312778110)
85. Cook PJ, Shergold JH. 1986 Proterozoic and Cambrian phosphorites-nature and origin. In *Proterozoic and Cambrian phosphorites* (eds PJ Cook, JH Shergold), pp. 369–386. Cambridge Univ. Press.
86. Parrish JT. 1987 Palaeo-upwelling and the distribution of organic-rich rocks. *Geol. Soc. Spec. Publ.* **26**, 199–205. (doi:10.1144/GSL.SP.1987.026.01.12)
87. Lu M, Lu YH, Ikejiri T, HoganCamp N, Sun Y, Wu Q, Carroll R, Cemen I, Pashin J. 2019 Geochemical

- 1
2
3
4
5
6
7
8
9
10
11
12
13
14
15
16
17
18
19
20
21
22
23
24
25
26
27
28
29
30
31
32
33
34
35
36
37
38
39
40
41
42
43
44
45
46
47
48
49
50
51
52
53
54
55
56
57
58
59
60
- Evidence of First Forestation in the Southernmost Euramerica from Upper Devonian (Famennian) Black Shales. *Sci. Rep.* **9**. (doi:10.1038/s41598-019-43993-y)
88. Marynowski L, Zatoń M, Rakociński M, Filipiak P, Kurkiewicz S, Pearce TJ. 2012 Deciphering the upper Famennian Hangenberg Black Shale depositional environments based on multi-proxy record. *Palaeogeogr. Palaeoclimatol. Palaeoecol.* **346–347**, 66–86. (doi:10.1016/j.palaeo.2012.05.02)
89. Buggisch W. 1991 The global Frasnian-Famennian »Kellwasser Event«. *Geol. Rundschau* **80**, 49–72. (doi:10.1007/BF01828767)
90. Friedman M. 2011 Parallel evolutionary trajectories underlie the origin of giant suspension-feeding whales and bony fishes. *Proc. R. Soc. B Biol. Sci.* **279**, 944–951. (doi:10.1098/rspb.2011.1381)
91. O’Keefe FR, Otero RA, Soto-Acuña S, O’gorman JP, Godfrey SJ, Chatterjee S. 2017 Cranial anatomy of *Morturneria* *seymourensis* from Antarctica, and the evolution of filter feeding in plesiosaurs of the Austral Late Cretaceous. *J. Vertebr. Paleontol.* **37**, e1347570. (doi:10.1080/02724634.2017.1347570)
92. Mark-Kurik E. 1992 The inferognathal in the Middle Devonian arthrodire *Homostius*. *Lethaia* **25**, 173–178. (doi:10.1111/j.1502-3931.1992.tb01382.x)

Tables

Table 1

Specimen Number	Species	Order	Scanning Institute
PIMUZ A/I 4716	Titanichthys termieri	Arthrodira	University of Zurich
PIMUZ A/I 4717	Tafilalichthys lavocati	Arthrodira	University of Zurich
CM6090	Dunkleosteus terrelli	Arthrodira	Cleveland Museum of Natural History
BMNH 1978.6.22.1	Cetorhinus maximus	Lamniformes	Natural History Museum, London
ZMA.PISC.108688	Heterodontus francisci	Heterodontiformes	Zoological Museum, Amsterdam
ERB 0932	Carcharodon carcharias	Lamniformes	ZNA hospital Antwerp
BMNH 1892.3.1.1	Balaenoptera musculus	Mysticeti	Natural History Museum, London
NMML-1850	Orcinus orca	Odontoceti	Idaho Museum of Natural History

Table 2

Species	Median			Mean		
	von Mises Stress	Maximum Principal Stress	Maximum Principal Strain	von Mises Stress	Maximum Principal Stress	Maximum Principal Strain
Titanichthys termieri	1.83	0.65	4.95E-05	2.72	1.43	9.43E-05
Dunkleosteus terrelli	0.68	0.31	2.17E-05	0.94	0.51	3.29E-05
Tafilalichthys lavocati	1.29	0.48	3.81E-05	1.79	0.94	6.14E-05
Cetorhinus maximus	0.78	0.36	2.52E-05	0.88	0.51	3.18E-05
Carcharodon carcharias	0.58	0.26	2.01E-05	0.82	0.48	3.03E-05
Heterodontus maximus	0.22	0.11	7.78E-06	0.30	0.18	1.11E-05
Balaenoptera musculus	9.32	3.71	2.90E-04	11.66	6.56	4.17E-04
Orcinus orca	1.78	0.62	5.27E-05	2.14	1.22	7.65E-05

Figures

1
2
3
4
5
6
7
8
9
10
11
12
13
14
15
16
17
18
19
20
21
22
23
24
25
26
27
28
29
30
31
32
33
34
35
36
37
38
39
40
41
42
43
44
45
46
47
48
49
50
51
52
53
54
55
56
57
58
59
60

1
2
3
4
5
6
7
8
9
10
11
12
13
14
15
16
17
18
19
20
21
22
23
24
25
26
27
28
29
30
31
32
33
34
35
36
37
38
39
40
41
42
43
44
45
46
47
48
49
50
51
52
53
54
55
56
57
58
59
60

Figure and table captions

Table 1

The specimens used in the study and the institutes in which they were scanned. Additional *Titanichthys* and *Tafilalichthys* specimens were observed at the University of Zurich to provide a more thorough insight into the species.

Table 2

Average elemental stress and strain values for the lower jaws of various species of placoderms, sharks and whales, calculated using finite element analysis. Both median and mean values are displayed. The unit for all values is MPa (megapascals).

Figure 1

Left inferognathal of *Titanichthys termieri* (PIMUZ A/I 4716), from the Southern Maïder basin, Morocco. The specimen is nearly complete, excluding the anteriormost tip. The inferognathal lacks both dentition and shearing surfaces. It has been glued together where fractures occurred. Photographed at the University of Zurich. Total length = 96 cm.

Figure 2

Inferognathals of *Tafilalichthys lavocati* (PIMUZ A/I 4717), from the Southern Maïder basin, Morocco. Photographed at the University of Zurich. Total length = 33cm.

Figure 3

Von Mises stress distributions in the lower jaws of selected placoderm, shark and whale species, calculated using finite element analysis (generally following the methodology of Snively *et al.* [34]).

Figure 4

Median von Mises stress values for each jaw finite element model. Bar colour corresponds with the potential ecological niche of each species.

Figure 5

Principal component analysis (PCA) visualising the variation in von Mises stress distribution between the lower jaw finite element models, as indicated by the intervals method [58]. Symbol colour is used to distinguish between clades: placoderm symbols are white and shark symbols are black. Shapes correspond with the potential ecological niche of each species. The percentage values on the axes indicate the variance explained by each principal co-ordinate. PC1 and PC2 cumulatively account for 90.6% of the total variance.

Appendix C

Response to Reviewers

We would like to thank the reviewers for their helpful feedback, and the time they spent to examine our manuscript. Specific responses are listed herein:

Reviewer 1

1. We agree with the reviewer's comment that utilising CT scans and treating all specimens equitably would have been preferable – perhaps this could be something to look into in future. Potentially we could also incorporate other megaplanktivores, like whale sharks, in the analysis.
2. We agree with the reviewer's comment that including *Leedsichthys* could have been massively enlightening. *Leedsichthys* specimens at Peterborough Museum were studied, but were judged to be insufficiently complete for production of a reliable 3D model.
3. As a response to the reviewer's comment, we have replaced every usage of the term "filter-feeding" with the term "suspension-feeding"; as well as altering the paragraph relating to the potential suspension-feeding method of *Titanichthys* and planktivorous pachycormids.
4. As a response to the reviewer's comment, we have mentioned the difficulty of directly tacking primary productivity, with a reference to Pyenson *et al.* (2016) – who used maximum body size as an indicator for primary productivity. (Introduction, paragraph 4).
5. As a response to the reviewer's comment, we have referenced the lateral head shake predatory strategy of *Carcharodon* (Martin *et al.*, 2005). (Discussion, section 1, paragraph 5).
6. We have incorporated all suggested grammatical improvements and rephrasings.

Reviewer 2

1. As a response to the reviewer's comment, we have changed "orbitals" to "orbits". (Introduction, paragraph 2).
2. With regards to the reviewer's question about gill raker robustness, it is our understanding that gill raker length, quantity and spacing are the primary diagnostic characteristics of planktivory (Liston, 2013). Robustness is likely to vary among suspension-feeders based upon the feeding style, with finer rakers well-suited for sieving and more robust rakers better able to handle higher buccal flow velocities.
3. As a response to the reviewer's comments, we have changed the terminology used regarding arthrodiran inferognathals as advised. (Materials and Methods, section 1, paragraph 2).

4. Whale shark specimens were not used as they were not available from the online repository that all other shark jaws were downloaded from (Kamminga *et al*, 2017). It was judged that maintaining methodological consistency in production of the 3D models was important for the validity of the intra-lineage comparisons, despite not being possible across all lineages.
5. As a response to the reviewer's comment, we have added an additional qualifier to the sentence regarding the phylogeny of the Mylostomatidae. (Conclusion, paragraph 3)